# Neurofunctional underpinnings of individual differences in visual episodic memory performance

Léonie Geissmann [1,2] ✉, David Coynel [1,2],
Andreas Papassotiropoulos [2,3,4,5] & Dominique J. F. de Quervain [1,2,4,5] ✉

Episodic memory, the ability to consciously recollect information and its context, varies substantially among individuals. While prior fMRI studies have identified certain brain regions linked to successful memory encoding at a group level, their role in explaining individual memory differences remains largely unexplored. Here, we analyze fMRI data of 1,498 adults participating in a picture encoding task in a single MRI scanner. We find that individual differences in responsivity of the hippocampus, orbitofrontal cortex, and posterior cingulate cortex account for individual variability in episodic memory performance. While these regions also emerge in our group-level analysis, other regions, predominantly within the lateral occipital cortex, are related to successful memory encoding but not to individual memory variation. Furthermore, our network-based approach reveals a link between the responsivity of nine functional connectivity networks and individual memory variability. Our work provides insights into the neurofunctional correlates of individual differences in visual episodic memory performance.

Human episodic memory (EM) refers to the conscious memory for personally experienced events within a particular spatio-temporal context[1]. It involves multiple brain systems during encoding, consolidation, and retrieval. The encoding phase relies on receiving information through sensory modalities and on cognitive integration, like content processing, attention attribution, and storage[2].

Extensive functional magnetic resonance imaging (fMRI) research has resulted in solid knowledge about neural activity related to successful EM encoding[2–6]. Most studies used the subsequent memory effect paradigm, in which one compares, in a voxel-wise manner on group-level data, brain activity during encoding stimuli later remembered with brain activity during encoding stimuli not later remembered. As a region-localizationist approach, it allows for pinpointing which brain regions play a role in successful memory encoding. A meta-analysis of visual EM reported subsequent memory effects in

many regions, including the left inferior frontal cortex, bilateral fusiform gyrus, bilateral medial temporal lobe, bilateral premotor cortex, bilateral occipital cortex, and bilateral posterior parietal cortex[2].

While such group-based voxel-level fMRI studies provide insight into the neurofunctional roles of brain regions common across a group of individuals for a given cognitive task, they allow no inferences about the substantial subject-to-subject variability and its association with inter-individual differences in cognitive performance[7]. In other words: It is unclear to what extent brain regions related to successful memory encoding also show variations in activity that explain why individuals differ in memory performance. While one could hypothesize that people with better memory performance also show higher activity in brain regions involved in successful encoding, previous studies counter this hypothesis. It has been shown, for example, that subjects with mild cognitive impairment, as compared to healthy controls,

[1]Division of Cognitive Neuroscience, Department of Biomedicine, University of Basel, Basel, Switzerland. [2]Research Cluster Molecular and Cognitive Neurosciences, University of Basel, Basel, Switzerland. [3]Division of Molecular Neuroscience, Department of Biomedicine, University of Basel, Basel, Switzerland. [4]University Psychiatric Clinics, University of Basel, Basel, Switzerland. [5]These authors jointly supervised this work: Andreas Papassotiropoulos, Dominique J. F de Quervain. ✉e-mail: leonie.geissmann@unibas.ch; dominique.dequervain@unibas.ch

show significantly greater hippocampal activation in an associative memory encoding task[8]. Further, it has been proposed that for a given performance level, subjects more skilled and more efficient in dealing with cognitive load would show less brain activation due to a higher neural efficiency[9,10].

In order to address inter-individual differences by investigating brain–behavior correlations, typical sample sizes of individual fMRI studies need to be scaled up substantially[11]. While much is known about the associations between inter-individual differences in cognitive performance and properties of brain structure[12–14] and between inter-individual variability in cognitive performance and resting-state activity[15–19], there are no large-scale studies investigating the relationship between task-based functional brain profiles and inter-individual differences in EM performance.

Even though the aforementioned group-based meta-analysis[2] was well-powered with 72 studies, sample sizes of the individual studies ranged from 12 to 25 participants. To the best of our knowledge, a substantially powered single-sample study (i.e., sample size well above 100 subjects) of subsequent memory effects with regard to EM is lacking. Comparing the results from our large single-center study with those of the meta-analysis serves to establish the validity and robustness of our study. Additionally, such alignment helps corroborate and strengthen the overall findings of the meta-analysis. Furthermore, most studies using the subsequent memory effect paradigm did not account for memorability. This phenomenon acknowledges that some items (pictures or words) are inherently more memorable than others due to specific features, such as semantics, esthetics, or emotional valence[20–22]. The lack of accounting for item memorability may pose a challenge to the interpretation of the previously reported subsequent memory effects as the portion of associated neural activity confounded by memorability may be substantial[23].

In the present study, we explored the neurofunctional basis of inter-individual differences in EM performance by including both a region-localizationist and a network-based approach. A distinctive feature of the human brain is its ability to flexibly reconfigure interactions within and between populations of neurons. These functional interactions, a term used to describe the co-activity of brain regions, indicate communication and coordination of brain activity[24,25]. Even in the absence of direct structural connections, abnormal activity at one region can cause dysfunction at other regions in a network[26]. Functional interactions are disregarded by the conventional region-localizationist approach, which assigns functional roles to separate brain regions and provides only a partial account of brain function[27–29]. Therefore, a more thorough understanding of the neural basis of inter-individual differences in EM can benefit from a network-based approach as a complement to the well-established region-localizationist voxel-based approach. We used independent component analysis (ICA) to extract the task-specific activity of functional connectivity networks (FCNs) for our network-based analysis. As brain activity differs between tasks and between populations of individuals, using this data-driven procedure instead of a template-based one circumvents violating the assumption of across-task- and across-population equality in the spatial topology of FCNs[30–32].

The current work relied on a large sample of healthy young adults ($n = 1498$) who participated in a single-center fMRI study on memory where one single MRI scanner was used for brain imaging. The subjects engaged in a picture encoding task inside the MRI scanner and a subsequent free recall task outside the scanner, during which they were instructed to describe in writing as many of the previously seen pictures as possible. This data allowed us to address the following questions: How does a classical group-based subsequent memory effect analysis of our data align with the findings from the meta-analysis on subsequent memory effects[2]? In what ways do the results of our subsequent memory effect analysis change when controlling for memorability? What results emerge from a voxel-based

brain–behavior correlation approach exploring brain activations related to inter-individual differences in memory performance, and how do these findings relate to the memorability-controlled subsequent memory effects? And finally, what results emerge from a network-based approach investigating the neural correlates of inter-individual differences in memory performance? Apart from advancing our basic understanding of the neural correlates contributing to the variability in EM performance among individuals, the present study could provide a foundation for future research aimed at relating individual biological characteristics to specific neural signals of EM.

## Results

### Behavior
We found large variability in performance in the free recall task, in which the subjects were asked to describe in writing as many pictures as possible that had been presented during the encoding task. The number of pictures freely recalled ranged from 5 to 55 ($M = 30.90$, $SD = 8.29$). No ceiling or floor effects were detected (Fig. 1).

### Subsequent memory effect: voxel-based
We first ran a classical group-based subsequent-memory effect analysis. We could replicate subsequent memory effects known from the literature[2]: in the left inferior frontal cortex, bilateral fusiform gyrus, bilateral medial temporal lobe, bilateral premotor cortex, bilateral occipital, and bilateral posterior parietal cortex. Moreover, there were subsequent memory effects located in the precuneus, lingual gyrus, cerebellum, thalamus, orbitofrontal cortex (OFC), ACC, and large parts of the frontal cortex, all bilaterally (Fig. 2, Fig. S1).

In line with the meta-analysis[2], we found negative subsequent memory effects in the central opercular cortex, Heschl's gyrus, precuneus, right frontal pole, right intracalcarine and lingual gyrus, juxtapositional junction, and the precentral gyrus (Fig. S2).

### Memorability-controlled subsequent memory effects
Next, we conducted the subsequent memory effect analysis while statistically controlling for memorability (see Methods). Whereas largely similar brain regions emerged, the extent of significant activations and corresponding $t$-values were smaller after controlling for memorability, notably in parietal, occipital, posterior cingulate, and cerebellar regions (Fig. 3, Fig. S3). Moreover, this analysis revealed subsequent memory effects that emerged only when controlling them for memorability, mainly located in the bilateral fusiform gyrus (Fig. S4).

We also detected memorability-controlled negative subsequent memory effects in regions similar to the classical negative subsequent memory effects (Fig. S5). Regions that did not show any significant

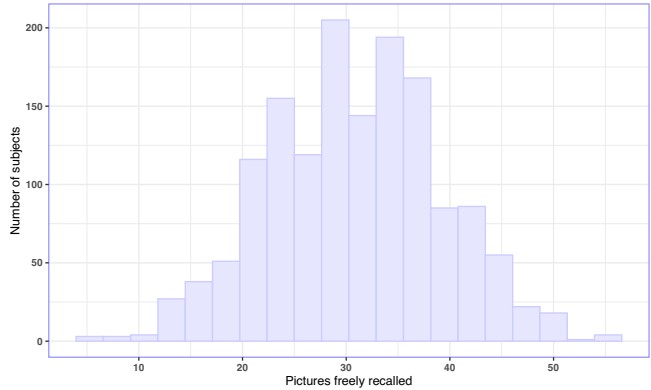

**Fig. 1 | Free recall performance.** The histogram illustrates free recall performance, defined as a number of pictures freely recalled ($M = 30.90$, $SD = 8.29$, range = 5 to 55; $n = 1498$). Source data are provided as a Source Data file.

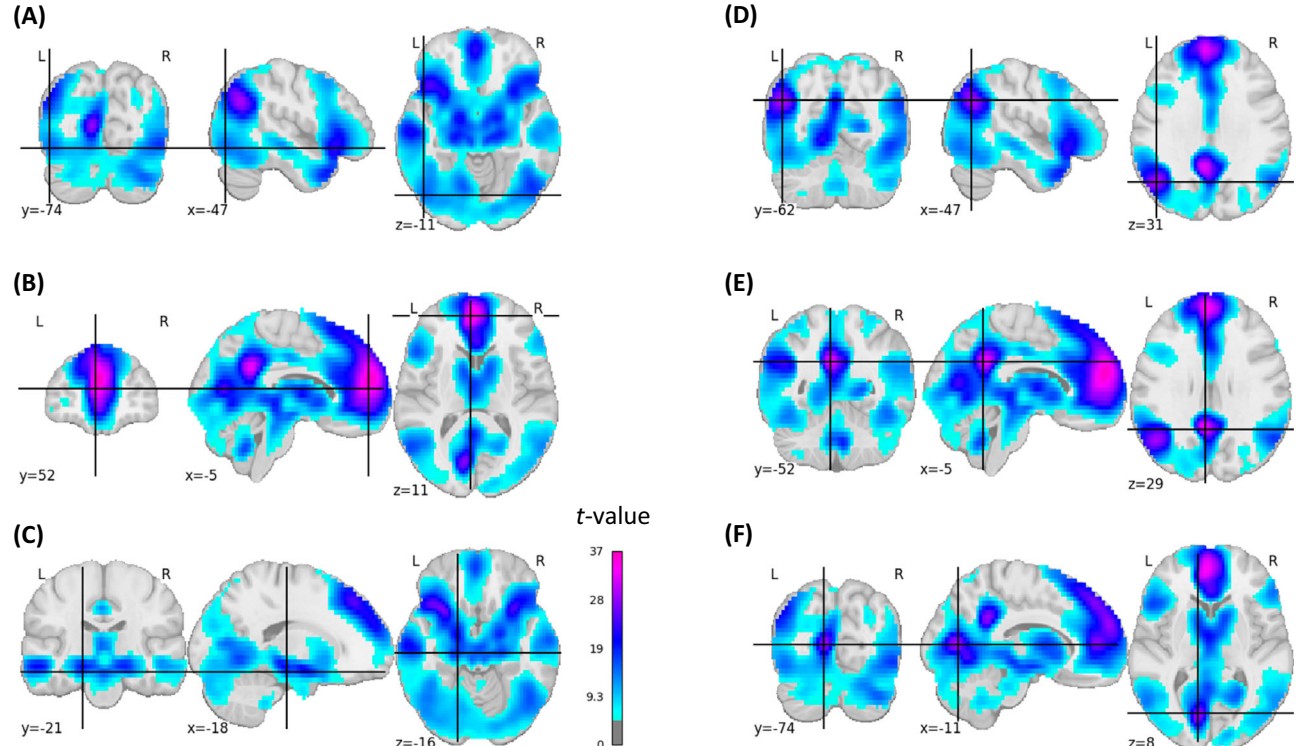

**Fig. 2 | Statistical brain map of the group-based positive subsequent memory effects.** For illustrative purposes, coordinates were placed in left-hemispheric brain regions: **A** inferior lateral occipital cortex ($t = 13.96$), **B** caudal anterior cingulate ($t = 35.28$), **C** hippocampus ($t = 17.15$), **D** superior lateral occipital cortex/angular gyrus ($t = 28.74$), **E** PCC ($t = 32.28$), **F** intracalcarine cortex ($t = 28.85$). The images are corrected for multiple comparisons at the whole brain level (two-sided $t$-test, $p$-FWE-corrected <0.05, $t$-FWE-corrected = 4.848).

negative effects when controlling for memorability were the intra-calcarine gyrus, lingual gyrus, and precentral gyrus.

For statistical brain maps representing the positive and negative effects of memorability on brain activation during encoding, see Figs. S6 and S7, respectively. The brain map illustrating the group-based positive memorability effects displays a robust activation pattern in memory-related regions, similar to those identified in the memorability-controlled subsequent memory effect analysis.

Additionally, we performed an analysis correcting the subsequent memory effects for picture arousal, one of the components of picture memorability. Again, we found a spatially similar activation pattern as for the classical subsequent memory effect, which was more focalized (i.e., including fewer voxels) and yielded lower $t$-values (Fig. S8).

### Brain–behavior correlations: voxel-based
At the voxel level, we detected positive brain–behavior correlations between brain responsivity to picture encoding and later EM performance in the left precuneus/left posterior cingulate cortex (PCC), medial OFC, superior frontal cortex (SFC), and bilaterally in the hippocampal formation (two-sided $p$-FWE-corrected < 0.05; 414 voxels; Fig. 4). There were no negative correlations.

### Reproducibility of brain–behavior correlations: voxel-based
To test the robustness of the voxel-based brain–behavior correlations based on the picture encoding contrast, we applied a resampling procedure (see Methods). This procedure consisted in estimating the effect size of brain–behavior correlations for sample sizes ranging from 26 to 1000 participants. Five thousand random samples were selected for each sample size. This analysis demonstrated a similar trend in effect sizes as the one reported in[33]: at small sample sizes, the association (i.e., brain–behavior correlation) was not reproducible and exhibited a lot of variability and sign changes. The effect size

converged at larger sample sizes and stabilized for sample sizes greater than 500 participants (Fig. 5).

### Comparison of voxel-based analyses
Next, we compared the memorability-corrected subsequent memory effects with the voxel-based brain–behavior correlations. Since the classical subsequent memory effects encompasses picture memorability-related activations that are similar across subjects, we used the memorability-corrected subsequent memory effects for this comparison. All brain regions with whole-brain-corrected brain–behavior correlations (Fig. 4) also demonstrated whole-brain-corrected memorability-controlled subsequent memory effects (Fig. 3). However, several memorability-controlled subsequent memory effects were located in brain regions that did not show brain–behavior correlations. To map out these regions more precisely, we examined the residuals of a regression analysis between the memorability-corrected subsequent memory effect analysis and the brain–behavior correlation analysis (Fig. S9). Regions where the brain–behavior correlations were lower than expected based on the memorability-corrected subsequent memory effects were mainly found in the left and right inferior and superior lateral occipital cortex (Fig. 6).

### Brain–behavior correlation of subsequent memory effects
Next, we explored whether inter-individual differences in subsequent memory effects might be related to memory performance. For this brain–behavior correlation, we constructed a model where the brain variable captured subsequent memory effects (see Methods). Whereas this analysis did not reveal any significant positive correlations, we observed negative correlations with EM performance in a few regions, most prominently in the lateral occipital cortex (Fig. S10), indicating that better performers show reduced subsequent memory effects in

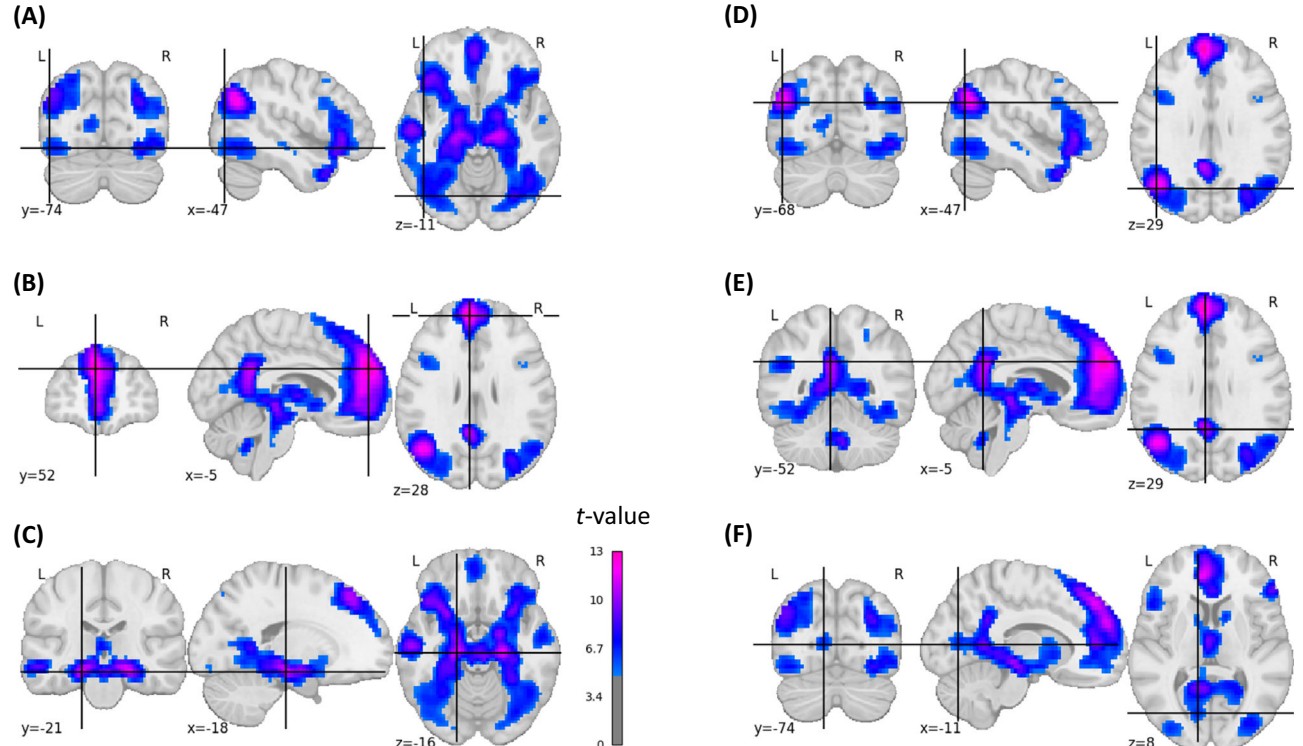

**Fig. 3 | Statistical brain map of the group-based memorability-controlled positive subsequent memory effects.** For illustrative purposes, coordinates were placed in left-hemispheric brain regions: **A** inferior lateral occipital cortex ($t = 6.68$), **B** superior frontal gyrus ($t = 13.43$), **C** hippocampus ($t = 11.40$), **D** superior lateral occipital cortex/angular gyrus ($t = 11.53$), **E** PCC ($t = 11.78$), **F** intracalcarine cortex ($t = 6.45$). The images are corrected for multiple comparisons at the whole-brain level (two-sided $t$-test, $p$-FWE-corrected < 0.05, $t$-FWE-corrected = 4.82).

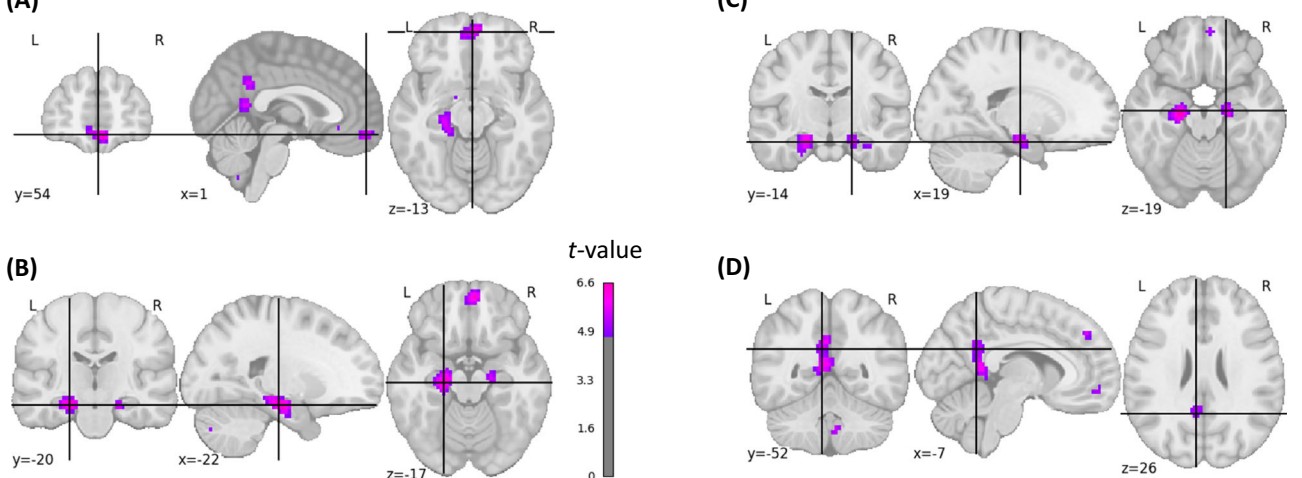

**Fig. 4 | Statistical brain map of brain–behavior correlations.** Shown are inter-individual correlations between brain responsivity during encoding and free recall performance using a voxel-based approach. Coordinates were positioned at the points of local maxima within the following brain regions: **A** medial OFC/frontal pole ($t = 5.61$), **B** hippocampus left ($t = 6.44$), **C** hippocampus right ($t = 5.85$), **D** PCC ($t = 5.15$). The images are corrected for multiple comparisons at the whole-brain level (two-sided $t$-test, $p$-FWE-corrected < 0.05, $t$-FWE-corrected = 4.799).

these regions as compared to lower performers. Of note, the lateral occipital cortex showed subsequent memory effects (Fig. 3) but a lack of brain–behavior correlations using the picture encoding contrast (Fig. 6).

### Network-based analyses: general

We used ICA to extract group-based FCNs in a data-driven manner. For the purpose of ICA decomposition and network validation, we split our

sample into two comparably large sub-samples (see Methods). This validation step involved comparing the solution of the ICA conducted in subsample 1 ($n = 590$) with the solution of the ICA conducted in subsample 2 ($n = 580$). Among 60 ICs (Fig. S11), between-sample spatial voxel correlations were high ($|r|_{max} > 0.6$) for 50 ICs, with a median of $r = 0.856$ (Table S1, Table S2, Fig. S12) and 25th and 75th quantiles at $r = 0.716$ and $0.915$, respectively. Next, we checked for the similarity of our task-based ICs with typical resting-state networks (RSN), as

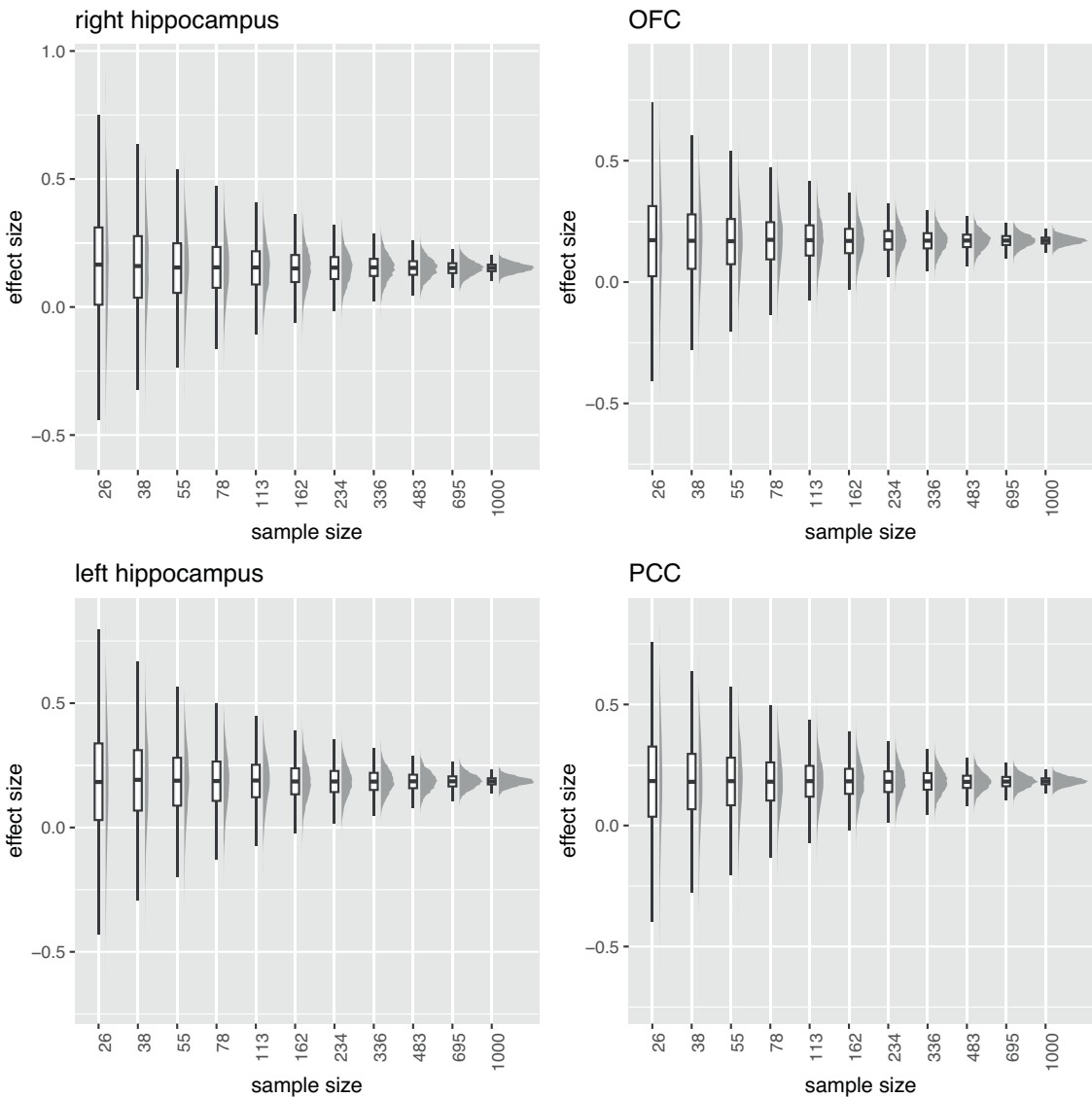

**Fig. 5 | Reproducibility of brain–behavior correlations.** Distribution (boxplot and histogram) of brain–behavior standardized effect sizes for various sample sizes ($n = 26, 38, 55, 78, 113, 162, 234, 336, 483, 695, 1000$) in the four clusters identified in the brain–behavior correlation analysis ($p$-FWE-corrected < 0.05, see Fig. 4). For every sample size, random participants were sampled 5000 times to compute the association. Boxes denote the 25th to 75th percentile and the median line. Whiskers extend 1.5 times the interquartile range from the edges of the box. Abbreviations: OFC orbitofrontal cortex, PCC posterior cingulate cortex. Source data are provided as a Source Data file.

previously done[34]. We did so by calculating cross-correlations between the ICs obtained from our sample and ten typical RSN, using a lenient and a more stringent threshold ($|r| > 0.1$ and $|r| > 0.2$, respectively). The mean number of matching RSNs per IC was $M_{lenient} = 2.083$ and $M_{stringent} = 1.5$ ($SD_{lenient} = 1.204$ and $SD_{stringent} = 0.682$). RSNs with high similarity to the ICs for which brain–behavior correlations were found (see below) were the cerebellum network, sensorimotor network, auditory network, and left the frontoparietal network in case of the stringent threshold, and additionally, the default mode network when considering the lenient threshold (Fig. S13, Fig. S14).

### Brain–behavior correlations: network-based
In this network-based analysis, we tested for links between network responsivity during encoding and memory performance. The responsivity of 9 ICs was associated with the number of pictures freely recalled (ICs 5, 6, 21, 29, 37, 42, 50, 52, 54), i.e., showed brain–behavior correlations (Fig. 7, Fig. S15, Fig. S16). Responsivity of IC 6 demonstrated a negative association with the number of pictures freely

recalled, while the other significant ICs showed a positive association. Variance explained by each of these IC's responsivity was small to medium[35], ranging from 3.5% to 5.8% (Table S3).

### Characterization of IC 5: cortico-cerebellar network
For the most part, IC 5 encompasses the right cerebellum as well as the left fronto-opercular, fronto-caudal, and fronto-rostral parts, temporal and parietal regions. The right cerebellum is important in cognitive processes like error processing, response inhibition, performance monitoring, memory, and emotional responding. Other brain regions of this IC are involved in memory integration, information binding, and planning[36–39]. Given its structural connections and functional implications, the cerebellum has been suggested as an add-on to the dorsal attention network[37], suggesting a cortico-cerebellar network.

### Characterization of IC 21: medial-frontoparietal network
IC 21 resembles not only the default mode network but also contains additional clusters. Anatomically, it includes the frontal pole, anterior-

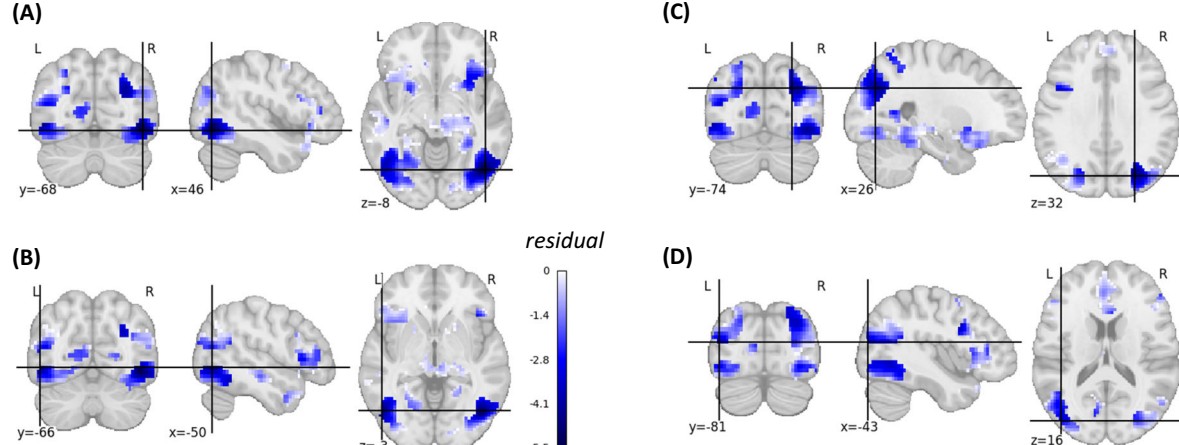

**Fig. 6 | Brain maps illustrating regions where brain–behavior correlations were lower than expected based on the memorability-corrected subsequent memory effects.** This figure illustrates the negative residuals (in blue) of a voxel-wise linear model where the predictors were *t*-values from the memorability-corrected subsequent memory effect analysis, and the outcomes were *t*-values obtained from the brain-behavior correlation analysis (see Methods). The figure is limited to voxels exhibiting a significant *p*-FWE-corrected memorability-corrected subsequent memory effect. **A** Inferior lateral occipital cortex right (residual = −5.53), **B** inferior lateral occipital cortex left (residual = −4.71), **C** superior lateral occipital cortex right (residual = −4.50), **D** superior lateral occipital cortex left (residual = −3.40). A linear regression model was used.

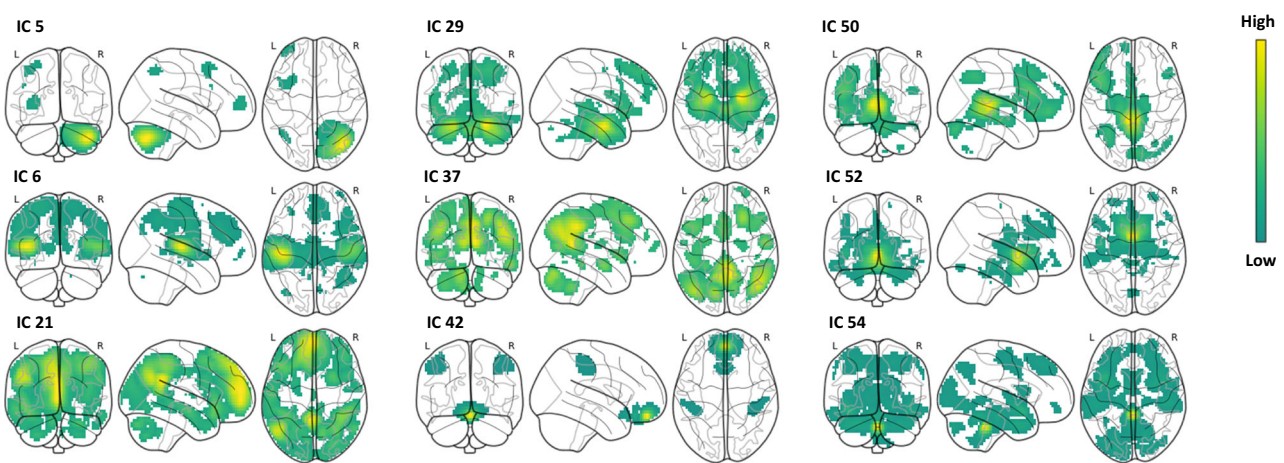

**Fig. 7 | The ICs with brain–behavior correlations.** *Z*-values run along a spectrum from yellow to dark green, respectively, with high to low values in glass brains. These values indicate the contribution of brain regions to the corresponding IC, irrespective of their link to behavior. Please note: IC 6 was negatively associated with the number of pictures freely recalled, while the other ICs were positively associated. For more detailed illustrations, please see Supplementary Material (Fig. S15).

medial OFC, superior frontal cortex, rostral anterior cingulate cortex (ACC), PCC, precuneus, isthmus cingulate cortex, occipital cortices, and angular gyrus. Among these regions' recognized functional roles are EM retrieval, higher-order cognition, visuo-spatial imagery, self-processing, and memory integration[31]. This network overlaps with IC 37 (see below).

### Characterization of IC 29: MTL network
Centered on the medial temporal lobe (MTL), IC 29 includes the parahippocampal gyrus, hippocampus, entorhinal cortex, and amygdala bilaterally. Additional brain regions are the brainstem, thalamus, and right cerebellum. These regions share fundamental roles in memory and emotion[40–42]. To a comparatively smaller extent, IC 29 includes non-neural areas.

### Characterization of IC 37: posterior default mode network
IC 37 resembles the previously described posterior component of the default mode network[31], overlapping with the ventral default mode network[43], both of which have been linked to self-directed processing

and EM. One cluster prominently covers the precuneus, posterior cingulate, intracalcarine, and lingual gyri, extending to the precentral and postcentral gyri. A left-hemispheric cluster is situated in the angular gyrus, middle temporal gyrus, supramarginal gyrus, and lateral occipital cortex. A similar albeit smaller cluster appears in the right hemisphere. IC 37 further includes parts of the left middle, superior, and frontal cerebellum.

### Characterization of IC 42: OFC network
IC42 is characterized by clusters in the medial OFC and in the bilateral postcentral gyrus, with a remarkably compact appearance. Covered brain regions are implicated in autobiographical memory recall, recollection of self-relevant information, emotion regulation, imagery, representational memory, and behavior-outcome-expectancy[44,45].

### Characterization of IC 50: extended left fronto-parietal network
IC 50 spans the superior frontal cortex, opercular cortex, lateral OFC, rostral and caudal frontal cortex, opercular cortex, inferior frontal cortex, cerebellum, precuneus, PCC, brainstem, thalamus, angular

gyrus, thereby sharing overlap with the left fronto-parietal network. Among the included brain regions' functions are executive function, affective and interoceptive processing, and memory integration[31,46]. Besides coverage of brain regions, IC 50 includes ventricular parts.

### Characterization of IC 52: ventral striatal-subcallosal network

IC 52 majorly covers the nucleus accumbens, caudate, and subcallosal cortex, extending to the OFC. The nucleus accumbens and OFC share both structural and functional connections[47]. The nucleus accumbens further has structural connections to the brainstem[48]. Associative appetitive and aversive learning is among the nucleus accumbens' functional implications[49,50]. The locus coeruleus is the primary source of norepinephrine and interacts with the nucleus accumbens[51], with implications in learning and memory, and with functional interactions with key brain regions for EM, such as the hippocampus[51,52], and the amygdala[53,54]. The subcallosal cortex, which interacts with cortical and subcortical regions, is functionally implicated in interoception, emotion, and memory, e.g., by gating hippocampal output to other cortices[55].

### Characterization of IC 54: insula-occipital-temporal network

Covering the superior lateral occipital cortex, precuneus, inferior and middle temporal gyrus, hippocampus, subcallosal cortex, precentral gyrus, insular cortex, brainstem, as well as ventricles, IC 54 has a fragmented appearance and partially overlaps with each of the other eight ICs with brain–behavior correlations (Fig. S17). Involvement of the insula, temporal gyri, and hippocampus may have fostered this IC to have brain–behavior correlations despite wide-ranging ventricular coverage. The insula, implicated in various cognitive, motor, somatosensory, and emotional functions[56], as a hub in the brain, is extensively connected across the brain.

### Characterization of IC 6: multi-modal integration network

IC 6 overlays sensory-motor and sensory-auditory areas. It includes the anterior and posterior cingulate cortices and the posterior insula. These brain regions, especially the posterior insula, have wide-spanning cognitive and sensory functions and wide-ranging structural connections, including cholinergic, dopaminergic, serotonergic, and noradrenergic systems[31,57]. Accordingly, we propose to label it a multi-modal integration network. IC 6 was the only network that was negatively associated with memory performance. It shows considerable overlap with memorability-controlled negative subsequent memory effects (Figs. S5 and S18).

All whole-brain-corrected voxel-based brain–behavior correlations were covered by one or more memory-related ICs (Table S4).

Our subsequent memory effect analysis, both corrected and uncorrected for memorability, revealed robust activations in the left and right inferior lateral occipital cortex. These regions were not only missing in the brain–behavior correlation analysis (Fig. 6), but they were also not included in any of the memory-related FCNs (Fig. S19).

## Discussion

The present single-center study in 1498 individuals allowed us to unravel both the neurofunctional underpinnings of successful EM encoding and the neurofunctional correlates of inter-individual differences in memory performance. With regard to the former, we replicated and extended the findings from a meta-analysis[2] on the neurofunctional underpinnings of successful memory, using the subsequent memory effect paradigm. With regard to the latter, using a brain–behavior correlation approach, we found both brain regions' and FCNs' responsivity to be associated with inter-individual differences in EM performance.

In line with numerous studies[2], the activations of the present subsequent memory effect analysis were located in the left inferior

frontal cortex, bilateral fusiform gyrus, bilateral MTL, bilateral posterior parietal cortex, bilateral occipital cortex, and bilateral premotor cortex. Regions not consistently reported previously included the precuneus, lingual gyrus, cerebellum, thalamus, OFC, ACC, and large parts of the frontal cortex. These additional findings are likely due to the high statistical power of our large single-center sample. While these additional findings apply to free recall of picture memory, it remains to be determined whether they also apply to EM involving other sensory modalities. In accordance with previous findings[2], we found negative subsequent memory effects in the superior temporal gyrus, pre- and postcentral gyrus, precuneus, lingual gyrus, insular gyrus, and superior frontal cortex.

This large sample size also offered the opportunity to decipher subsequent memory effects while statistically controlling for the systematic variation in picture memorability. This analysis yielded a spatially similar pattern with more focalized effects (i.e., including fewer voxels) and with lower activations overall. This finding is in line with the results of a study indicating that memorability confounds and overestimates subsequent memory effects to a considerable degree[23]. As an exception to this observed pattern, our analysis revealed subsequent memory effects, mainly located in the bilateral fusiform gyrus, that emerged only when controlling them for memorability.

The voxel-based brain–behavior correlation approach revealed that inter-individual differences in memory performance were associated with the responsivity of voxels in the left precuneus/left PCC, OFC, and bilaterally in the hippocampal formation. Each of these brain regions that contributed to explaining individual differences in EM performance was also related to memorability-controlled successful memory encoding. In contrast, there were several brain regions related to memorability-controlled successful memory encoding that did not explain inter-individual differences in EM performance, as evidenced by the lack of correlation between brain responsivity during encoding and free recall performance. These regions were mainly located in the lateral occipital cortex. Importantly, the left and right inferior lateral occipital cortex were also not part of any of the FCNs correlated with memory performance. This area, which belongs to the visual associative cortex, has been linked to the initial encoding and subsequent memory of visual stimuli[2]. Moreover, evidence from transcranial magnetic stimulation studies supports a causal role of this region in visual memory[58,59]. Thus, while the lateral occipital cortex appears to play a role in successful visual memory encoding, inter-individual differences in encoding-related brain activation in this region did not contribute to memory variability in the present study.

In the network-based brain–behavior correlation analysis, we found network responsivity during the encoding of nine FCNs to be associated with the later free recall. The nine ICs only partly match previously described FCNs or RSNs, in line with state-specific and task-specific flexibility in network configuration[60]. Labels for this set of ICs with brain–behavior correlations were selected based on previous literature and the ICs' spatial representations in the brain.

Among the FCNs for which higher responsivity was associated with improved recall is the cortico-cerebellar network (IC 5). Its brain regions are implicated in visual working memory, emotion, visual attention, executive functions, memory, cortico-striatal plasticity, and the conscious representation of memory[36–39]. IC 21 consists of regions in the frontal pole, OFC, superior frontal cortex, ACC, PCC, precuneus, isthmus CC, occipital cortex, lingual gyrus, parahippocampal gyrus, temporal gyrus, and opercular cortex. Given the overlap with the default mode network, this network is presumably involved in internally-oriented processing and memory. The default mode network's setup is assumed to be task-dependent and may consist of multiple subnetworks[61–63]. Accordingly, IC 37, the posterior default mode network, was also related to EM performance in our study. IC 29 consists of MTL regions, including the amygdala, hippocampus, parahippocampal gyrus, entorhinal cortex, and brainstem, but also

ventricular regions. The MTL is well-known for its role in memory[40–42]. To the best of our knowledge, IC 42 has not been reported as an FCN so far. It consists of the medial OFC and postcentral gyrus. The OFC is important for outcome expectancy, representational memory, impulsivity, and decision making[44,45], and has functional connections to the default mode network, limbic regions, hippocampus, striatum, and thalamus. As opposed to IC 42's compact appearance, IC 50 consists of a large number of brain regions, that is, the superior frontal cortex, opercular cortex, right inferior frontal cortex, left lateral OFC, opercular cortex, inferior and caudal frontal cortex, cerebellum, precuneus, PCC, brainstem, and thalamus. It overlaps with the left frontoparietal network, which is implicated in language, executive function, inhibitory control, pain, and sensory processing[46]. IC 52 largely covers interconnected ventral-striatal regions, including the nucleus accumbens, subcallosal cortex, and brainstem. With major roles in attention and arousal, they are implicated in learning and memory[51]. In contrast to the other eight ICs, IC 54 stands out by combining gray matter and prominent spatial characteristics indicative of noise components. The latter include a fragmented appearance, large involvement of ventricles, and ring-like stripes near the edges of the field of view[64]. Involvement of the insula, temporal gyri and hippocampus may have fostered this IC to have brain–behavior correlations despite these noise components.

Network responsivity of IC 6 was negatively associated with memory performance, i.e., the stronger this FCN responds to stimuli, the fewer pictures were remembered later. IC 6 consists of extensively connected regions, such as sensory-motor, and sensory-auditory areas, ACC, PCC, juxtapositional cortex, and posterior insula. The insula is important for interoception, emotions, memory, sensory processing and integration, and attention[57,65]. The involvement of the insula in IC 6 could, therefore, be seen as beneficial for memory. However, the involvement of sensory-auditory areas could reflect auditory processing in an environment with high-volume auditory input (i.e., the auditory noise from the rapidly switching gradients in the MRI environment). It is possible that processing and integrating auditory signals may interfere with the visual memory task and consequently result in lower memory performance. In accordance with its direction of effect on memory performance, IC 6 spatially coincides with the negative subsequent memory effects.

It is noteworthy that almost all ICs with brain–behavior correlations were largely included in the brain regions whose brain activity during encoding, on a group level, has been found to be associated with successful recollection (i.e., memorability-corrected subsequent memory effects). Outstanding in this regard is the cortico-cerebellar network (IC 5) with involvement of the right cerebellar hemisphere that was also not detected by the voxel-based brain–behavior correlation approach. Since the cerebellum does not have the same microscopic structure as the cerebral cortex[66], its functional specialization may be better represented in variations in anatomical connectivity rather than variations in local microstructure[66,67]. Indeed, cerebellar FCNs have been shown to reconfigure during states of cognitive tasks compared to resting conditions and to be highly flexible depending on the cognitive task[61], highlighting the benefit of using FCNs based on the functional architecture present during a specific task to best capture associations with a relevant behavioral phenotype.

A particular feature of our study lies in the combined use of an approach that averages brain activity across individuals and an approach that addresses inter-individual differences. While the former, in order to explain a shared basic mechanism, wishes to minimize inter-individual variance by group averaging, the latter wishes to maximize variability to describe the association between behavior and neural underpinnings and requires large samples[11]. The large sample size and the fact that all subjects were investigated in the same scanner in our study is therefore beneficial with regards to statistical power and suitability for the inter-individual approach used here. Our resampling

analysis demonstrated that even within our homogeneous sample, we require between 500 and 1000 subjects to yield robust effects. This finding aligns with a recent publication, which asserts that reproducible brain-wide association studies require thousands of individuals[33].

In conclusion, our study identifies the key brain regions and networks related to individual differences in visual EM performance. Notably, we found that certain regions, pivotal at the group level, do not correlate with individual performance. These insights bear significant implications for research striving to link individual neurofunctional signals with psychological traits or with genetic, epigenetic, or metabolomic profiles. Research of this nature would benefit from the selection of neurofunctional signals that are related to individual differences in memory performance rather than those that emerge from group-level analyses.

## Methods

### Experimental design

**Sample and study.** Data presented in this paper comes from a large single-center study aimed at uncovering neurobiological mechanisms underlying EM and working memory by combining genetic, behavioral, eye-tracking, and neuroimaging data[68,69]. The sample (complete data for $n = 1498$; 930 females) consists of healthy young adults aged 18–35 (25th percentile = 20, 75th percentile = 24; $M = 22.44$, SD = 3.31). The subjects were free of any lifetime neurological or psychiatric illness and did not take any medication at the time of the experiment (except hormonal contraceptives). All subjects gave written informed consent before participation in the study. The ethics committee of the Canton of Basel, Switzerland, approved the study protocol. After a short introduction, subjects were guided inside the MRI scanner to perform one run (21 min) of a picture encoding task, followed by a separate working memory task, whilst fMRI data was being collected. Then followed an unannounced free recall task outside the scanner. Participant compensation was CHF 25 per hour of study participation.

**Behavioral tasks: encoding task.** Seventy-two pictures selected from the International Affective Picture System (IAPS)[70] were used for the EM encoding task, equally distributed between neutral, negative, and positive valence categories. Eight neutral pictures were selected from an in-house standardized picture set in order to equate the picture set for visual complexity and content (e.g., human presence). Examples of pictures are as follows: erotica, sports, and appealing animals for the positive valence; bodily injury, snake, and attack scenes for the negative valence; and finally, neutral faces, household objects, and buildings for the neutral condition. Additionally, intermingled in between the IAPS pictures, 24 scrambled pictures with 24 distinct, simple geometrical figures (rectangle or ellipse of different sizes and orientations)[71] were presented in such a way that a maximum of two IAPS pictures were presented in succession. The scrambled background, on which a simple geometrical figure was presented, was created using Adobe Photoshop CS3 (©2007 Adobe Systems Incorporated). This background was composed of the IAPS pictures positioned one next to another, edited with a distortion and crystal filter in such a way that the motives were no longer perceivable. All IAPS pictures and scrambled pictures were presented in succession, following the above-mentioned rule. There was no repetition of the scrambled pictures with geometrical figures. The IAPS pictures were presented for 2.5 s in a quasi-randomized order so that at maximum, four pictures of the same valence category occurred consecutively. A fixation cross appeared on the screen for 500 ms before each picture presentation. The stimulus onset time was jittered within 3 s (1 repetition time [TR]) per valence category with regard to the scan onset. Consequently, trials were separated by a variable intertrial period of 9 to 12 s (jitter). During the intertrial period, participants rated the IAPS pictures according to valence (negative, neutral, or positive) and arousal (low,

middle, or high) on a 3-point scale (self-assessment manikin) by pressing a button with their dominant hand. For geometrical figures, which were presented on top of the scrambled background, participants rated their form (vertical, symmetrical, or horizontal) and size (large, medium, or small) during the intertrial period. Thus, each trial lasted between 12 s and 15 s (Fig. S20). We included 4 additional IAPS pictures, 2 at the beginning and 2 at the end of the task. These pictures were used as primacy and recency pictures, respectively, as these tend to be better remembered because of their position. Primacy and recency pictures were the same in all subjects and were not considered in the memory recall test. The software Presentation (Neurobehavioral Systems, Inc., Berkeley, CA; https://www.neurobs.com) was used for the presentation of the material within the scanner, using MR-compatible LCD goggles (VisualSystem, NordicNeuroLab). Subjects were kept uninformed about the upcoming free recall task.

**Behavioral tasks: free recall task**. In the free recall task, subjects were instructed to describe in writing as many of the previously seen pictures as possible. There was no time limit for completion. Due to expected presentation order effects, primacy and recency IAPS pictures were excluded from the analysis of free recall performance. Three independent raters were responsible for the scoring: two of the raters independently rated a picture as either recalled or not based on the participants' written picture description. The third rater then took a final decision in the case of differences in scoring between raters 1 and 2. Inter-rater reliability of the two raters was >98%. The amount of correctly recalled pictures, excluding the primacy and recency pictures, was our behavioral variable of interest.

## fMRI data acquisition

**MRI scanning parameters**. All functional and structural images were acquired on the same Siemens Magnetom Verio 3 T whole-body MR scanner equipped with a 12-channel head coil. Blood oxygen level-dependent fMRI was acquired using a single-shot echoplanar sequence along with generalized auto-calibrating partially parallel acquisition (GRAPPA), using the following parameters: echo time (TE) = 25 ms, field of view (FOV) = 22 cm, acquisition matrix = 80 × 80 (interpolated to 128 × 128, voxel size = 2.75 × 2.75 × 4 mm$^3$), acceleration factor = 2. We used an ascending interleaved sequence with a repetition time (TR) = 3000 ms (alpha = 82°), measuring 32 contiguous axial slices that were placed along the anterior-posterior commissure plane based on a midsagittal scout image.

A magnetization-prepared rapid acquisition gradient echo T1-weighted image was acquired using the following parameters: TR = 2000 ms, TE = 3.37 ms, TI = 1000 ms, flip angle = 8°, 176 slices, FOV = 256 mm, voxel size = 1 mm$^3$.

## Statistical analyses

**fMRI preprocessing**. fMRI data was preprocessed using SPM12 (Statistical Parametric Mapping, Wellcome Trust Center for Neuroimaging; http://www.fil.ion.ucl.ac.uk/spm/) implemented in MATLAB R2016b (MathWorks).

Volumes were slice-time corrected to the first slice (acquired at TR/2), realigned using the 'register to mean' option, and co-registered to the anatomical image by applying a normalized mutual information 3-D rigid-body transformation. Successful co-registration was visually verified for each subject. Subject-to-template normalization was done using DARTEL[72], which allows registration to both cortical and subcortical regions and has been shown to perform well in volume-based alignment[73]. Normalization incorporated the following four steps: (1) Structural images of each subject were segmented using the 'New Segment' procedure in SPM12. (2) The resulting gray and white matter images were used to derive a study-specific group template. The template was computed from a subgroup of 1000 subjects, which were part of the subjects included in the present study. (3) An affine

transformation was applied to map the group template to MNI space. (4) Subject-to-template and template-to-MNI transformations were combined to map the functional images to MNI space. The functional images were smoothed with an isotropic 8 mm full-width at half-maximum (FWHM) Gaussian filter.

Normalized functional images were masked using information from their respective T1 anatomical image as follows. At first, the three-tissue classification probability maps of the "Segment" procedure (gray matter, white matter, and CSF) were summed to define the brain mask. This mask was binarized, dilated and eroded with a 3 × 3 × 3 voxels kernel using fslmaths (FSL) to fill in potential small holes. The previously computed DARTEL flow field was used to normalize the brain mask to MNI space at the spatial resolution of the functional images. The resulting non-binary mask was thresholded at 50% and applied to the normalized functional images. Consequently, the implicit intensity-based masking threshold usually employed to compute a brain mask from the functional data during the first level specification (spm_get_defaults('mask.thresh'), by default fixed at 0.8) was not needed any longer and set to a lower value of 0.05.

Each participant's anatomical image was further automatically segmented into cortical and subcortical structures using FreeSurfer (v. 4.5)[74] Labeling of the cortical gyri was based on the Desikan–Killiany atlas[75] yielding 35 cortical and seven subcortical regions per hemisphere. Segmentations of cortical and subcortical structures were used to build a population-average probabilistic anatomical atlas based on data from the participants used to build the study-specific template. Individual segmented anatomical images were normalized to the study-specific anatomical template space using the participants' previously computed warp field and were affine-registered to the MNI space. Nearest-neighbor interpolation was applied to preserve the labeling of the different structures. The normalized segmentations were finally averaged across participants to create a population-average probabilistic atlas. Each voxel of the template could consequently be assigned a probability of belonging to a given anatomical structure. This population-average probabilistic atlas was used to report the anatomical location of coordinates and ROIs. Percentages per coordinate denoted the population-average probability of an anatomical label.

**Subsequent memory effects**. As in typical subsequent memory effect analyses, we proceeded in a standard hierarchical GLM implemented in SPM12. First-level analyses were conducted to identify subject-specific memory-related activations. Regressors modeling the onsets and duration of stimulus events were convolved with a canonical hemodynamic response function (HRF). More precisely, the model comprised regressors for button presses modeled as stick/delta functions, picture presentations (IAPS pictures later recalled, IAPS pictures later not recalled, primacy and recency) modeled with an epoch/boxcar function (duration: 2.5 s), and rating scales modeled with an epoch/boxcar function of variable duration (depending on when the subsequent button press occurred). Serial correlations were removed using a first-order autoregressive model, and a high-pass filter (128 s) was applied to remove low-frequency noise. Six movement parameters were also entered as nuisance covariates. The contrast estimate "IAPS pictures later recalled−IAPS pictures later not recalled" was computed for every subject and used as input for the following group-level analyses: subsequent memory effects and brain−behavior correlations.

The group-level analysis considered the average activation for the "IAPS pictures later recalled−IAPS pictures later not recalled" contrast and was implemented in MRTools' GLM Flex Fast2 (https://habs.mgh.harvard.edu/researchers/data-tools/glm-flex-fast2/). The model included age, sex, and batch effects (two MR gradient changes, one MR software upgrade, and one of two rooms in which subjects completed the free recall task) as additional regressors. Whole-brain two-sided

FWE correction for multiple comparisons was applied at a threshold of $p < 0.05$, with a minimum cluster size of 20 voxels.

**Subsequent memory effects controlled for memorability.** Subsequent memory effects analyses considering picture memorability were also conducted. Picture memorability was defined as the average free recall score of a picture; over 1739 subjects performed this free recall task, including those from this study. First-level models were run, including the following regressors: IAPS pictures presentation, geometrical figures presentation, rating scales presentation, button presses, and 6 movement parameters (not convolved with the HRF). Additionally, two parametric regressors (PM) were added for the "IAPS pictures" regressor: (1) memorability-PM: picture memorability score of each picture; (2) subjective memory-PM: whether the picture was remembered or not. The PM regressors are orthogonalized with respect to the unmodulated regressor, and the second PM is orthogonalized with respect to the first one. The interpretation for the unmodulated regressor is the mean activation across trials. The memory-PM regressor captures memory-related variability of the BOLD response that is not explained by (a) the canonical HRF (mean activation) and (b) variability due to memorability effects.

The group-level analyses considered the average activations for the memory-PM and memorability-PM regressors. The model included age, sex, and batch effects (two MR gradient changes, one MR software upgrade, and one of two rooms in which subjects completed the free recall task) as additional regressors. Whole-brain two-sided FWE correction for multiple comparisons was applied at a threshold of $p < 0.05$, with a minimum cluster size of 20 voxels. The group-level analysis regarding the memorability-PM regressors is described in the Supplementary materials.

**Subsequent memory effects controlled for arousal.** Akin to the analysis investigating subsequent memory effects controlled for memorability, we investigated how picture arousal affects subsequent memory. Picture arousal was defined as the average arousal score of a picture, averaged over 1739 subjects that performed this encoding task, including those from this study. A similar parametric modulation analysis was setup, using the following two parametric regressors: (1) arousal-PM: the picture arousal score of each picture; (2) subjective memory-PM: whether the picture was remembered or not. In this context, the memory-PM regressor captures memory-related variability of the BOLD response that is not explained by (a) the canonical HRF (mean activation) and (b) variability due to arousal effects.

The group-level analysis considered the average activation for the memory-PM regressor. The model included age, sex, and batch effects (two MR gradient changes, one MR software upgrade, and one of two rooms in which subjects completed the free recall task) as additional regressors. Whole-brain two-sided FWE correction for multiple comparisons was applied at a threshold of $p < 0.05$, with a minimum cluster size of 20 voxels.

**Brain–behavior correlations.** Brain–behavior correlations were investigated based on the following first-level contrasts: picture-encoding activations and subsequent memory effects. To identify picture-encoding subject-specific activations, the following first-level analyses were conducted: the model comprised regressors for button presses modeled as stick/delta functions, picture presentations (IAPS pictures, scrambled pictures, primacy, and recency) modeled with an epoch/boxcar function (duration: 2.5 s), and rating scales modeled with an epoch/boxcar function of variable duration (depending on when the subsequent button press occurred). Serial correlations were removed using a first-order autoregressive model, and a high-pass filter (128 s) was applied to remove low-frequency noise. Six movement parameters were also entered as nuisance covariates. The contrast estimate "IAPS pictures−scrambled pictures" was computed for every subject and used as input for the group-level brain–behavior correlation analysis (the average estimated standardized beta over all trials). This contrast yields neural activity related to picture viewing and contains activations in brain regions typically involved in successful memory encoding[76].

The brain–behavior correlation analyses investigated the relationship between individual contrasts ("IAPS pictures−scrambled figures" or "IAPS pictures later recalled−IAPS pictures later not recalled") and free recall memory performance by means of linear models. The models included age, sex, and batch effects (two MR gradient changes, one MR software upgrade, and one of two rooms in which subjects completed the free recall task) as additional regressors. Whole-brain two-sided FWE correction for multiple comparisons was applied at a threshold of $p < 0.05$, with a minimum cluster size of 20 voxels.

**Reproducibility of brain–behavior correlations.** As recently reported in[33], robust brain–behavior correlation analyses require sample sizes much larger than in classical mass-univariate voxel-based analyses. We seized the opportunity to investigate whether a similar pattern was observed in our data. We extracted the mean picture-encoding activation in the 4 largest clusters that had a significant brain–behavior correlation in the whole sample ($p$-FWE-corrected <0.05 and cluster size of at least 20 voxels). We specified linear models to investigate the relationship between mean brain activation and memory performance, including age, sex, and batches as covariates. The output variable of interest for these analyses was the standardized effect size, akin to the correlation coefficient used in[33]. We randomly selected participants from the whole cohort at various sample sizes (logarithmically spaced samples $n = 26, 38, 55, 78, 113, 162, 234, 336, 483, 695, 1000$). For every sample size, participants were randomly selected 5000 times. The distribution of effect sizes was plotted for every sample size in the 4 regions of interest. The plots were created using ggdist[77].

**Voxel-based approaches: comparison of the memorability-controlled subsequent memory effects and the voxel-based brain–behavior correlations.** Strong brain–behavior correlations are expected to occur in memory-related regions, i.e., regions that exhibit a strong memorability-corrected subsequent memory effect. In order to quantify the strength of this relationship, we compared the group-level $t$-values of the two analyses across the whole brain. A linear model was specified, with all voxels' memorability-corrected subsequent memory effect $t$-values as the predictor and brain–behavior correlation $t$-values as the outcome variable. We then extracted the residuals of the linear model in order to graphically illustrate regional deviations from the general whole-brain pattern. Hence, we obtained one residual value for each voxel in the whole brain. Positive residuals represent regions where the brain–behavior correlation is as strong or stronger than predicted based on the memorability-corrected subsequent memory effect $t$-values, while negative residuals represent regions where the brain–behavior correlation is weaker than predicted. The corresponding brain images depict the residuals only in voxels with significant memorability-corrected subsequent memory effects.

**Network extraction and validation in two subsamples: ICA.** Using group probabilistic spatial ICA[78], we first decomposed brain activity during encoding into 60 spatially independent components (IC). This number of ICs yielded an optimal balance between dimensionality reduction and loss of information. ICA input data consists of all subjects' data concatenated in the time dimension (60,638 voxels × 420-time points of $n$ subjects). Importantly, the algorithm does not give any information about the task but instead separates signals into independent spatial sources that together explain brain activity in a purely data-driven manner.

The resulting spatial maps were thresholded using an alternative hypothesis test based on fitting a mixture model to the distribution of

voxel intensities within spatial maps using the default parameters (https://fsl.fmrib.ox.ac.uk/fsl/fslwiki/MELODIC#MELODIC_report_output)[79].

Network extraction was done for two subsamples independently, consisting of 590 and 580 subjects each (subsamples 1 and 2, respectively). Network extraction calculations were performed on sciCORE (http://scicore.unibas.ch/) scientific computing center at the University of Basel, Switzerland, on a single node with 128 GB of RAM. Due to characteristics inherent to FLS's MELODICS, the job was running on a single core. Based on these computational limitations, this analysis did not use the full sample size. This allowed us to validate the decomposition in subsample 1 and to proceed with replicable networks only. For each of both subsamples' decompositions, we extracted all unthresholded IC's voxel loadings and cross-correlated them with all IC's voxel loadings of the other sample. ICs with $|r|_{max} \geq 0.7$ were regarded as replicable. ICs with $|r|_{max} \geq 0.6$ and $|r|_{max} < 0.7$ were visually inspected to make a judgment on their replicability. All other ICs were treated as insufficiently replicable and were therefore not considered for interpretation. The value $|r|_{max}$ describes the maximum correlation value of an IC of subsample 1 with any IC of subsample 2, i.e., regardless of the number of matches passing the threshold. Corresponding figures were created in the R environment[80] (v. 4.1.2) with the library ggplot2 (v. 3.4.0)[81].

**Network time course calculation in all subjects: dual regression.** The next step was to get subject-specific time courses for the 60 ICs obtained from subsample 1 running dual regression in FSL v.5.0.9[78]. The set of spatial maps from the group-average analysis was used to generate subject-specific versions of the spatial maps, and associated time-series, using dual regression[82,83]. First, for each subject, the group-average set of spatial maps is regressed (as spatial regressors in a multiple regression) into the subject's 4D space-time dataset. This results in a set of subject-specific time series, one per group-level spatial map, for a final sample size of $n = 1485$. Thirteen subjects were not included due to the non-availability of dual regression data at the time point of data analysis.

**Network responsivity.** Network responsivity analyses were implemented in R (v. 4.1.2)[80]. The R library dplyr was used to filter and merge data (v. 1.0.10)[84]. Functional modulation of each component for each subject was estimated in a first-level analysis, including the following regressors: IAPS pictures, geometrical figures, primacy and recency pictures, stimuli rating, button press, and six movement parameters. The task regressors were convolved with the hemodynamic function for the voxel-based analyses. The dependent variable was each IC's subject-specific time course. The difference between IAPS pictures and geometrical figures estimates (the average estimated standardized beta over all trials) was used as a measure of task-related functional responsivity of each IC[85]. The R library nlme (v. 3.1–153)[86] was used for the first-level analysis.

Those contrast estimates were used to examine their relationship with inter-individual differences in memory by means of linear models. Each model included all subjects' contrasts as the independent variable of interest, the number of correctly recalled pictures as the dependent variable, and the covariates sex, age, and batch effects (two MR gradient changes, one MR software upgrade, one of two rooms in which subjects completed the free recall task). All results were corrected for multiple comparisons to reduce the burden of false positives: a Bonferroni correction was applied by dividing the statistical threshold by the number of ICs, resulting in a threshold of $p < 8.33e-04$ (0.05/60).

**Network characterization.** Anatomical labeling of the ICs was based on the FreeSurfer Desikan–Killiany atlas labels described in *fMRI preprocessing*.

The spatial maps calculated in FLS's MELODIC are the projections of the data onto the estimate of the unmixing matrix. This data, per default, has been de-meaned in time and space and normalized by the voxel-wise standard deviation (i.e., pre-processed by MELODIC). The individual spatial maps result from multiple regression rather than being correlation maps of the voxels' time courses. The default thresholding approach involves steps of inferential calculations. We use the thresholds calculated by MELODIC for all IC-based analyses. For the purpose of descriptive characterization, we applied arbitrarily selected thresholds (i.e., $z = \{3,4,5\}$) to provide a notion of the contribution of individual voxels to the IC.

**Network characterization: similarity to RSNs.** As done previously[34], we quantified the similarity of our task-related ICs to a set of 10 resting-state templates, which have been robustly detected in a number of independent studies[31,87,88], available on http://www.fmrib.ox.ac.uk/dazasezs/brainmap+rsns/ (retrieved 07/07/2016), described in. These template RSNs circumscribe three visual networks (medial, occipital pole, lateral visual areas; 1–3), the default mode network (DMN), a cerebellum network (CN), the sensorimotor network (SMN), auditory network (ADT), executive control network (ECN) and left/right fronto-parietal networks (LFPN, RFPN). We identified the template RSNs that had the highest spatial correlation with our task-based ICs using FSL's spatial cross-correlation function. We used the R library networkD3 to create Fig. S13B and Fig. S14 (v. 0.4)[89].

**Network characterization: similarity to the subsequent memory effect.** The procedure was the same as the one for the calculation of similarity between the brain–behavior correlations from the voxel-based and network-based approaches (see above).

**Network characterization: visual inspection and characterization of the independent components with brain–behavior correlations.** ICA separates the data into a set of spatial maps that together compose the whole-brain data[46,90]. Due to its ability to simultaneously denoise as well as capture variances in the BOLD signal[60], careful visual inspection of the ICs is a critical step to reap its full benefits. We carefully visually inspected the ICs such as to be sure to draw valid conclusions based on the findings from the network-based brain–behavior correlations, keeping in mind the drawbacks and benefits of the data-driven approach of ICA. Examples of noise components are strong loadings in the ventricular system and movement-related ring artifacts at the periphery of the cortex. We further provide detailed descriptions of which brain regions are included in the ICs and what their implications are.

**Brainmaps: figure creation.** Nifti images in R were created utilizing functions from the R-package oro.nifti (v. 0.11.4)[91].

Figures illustrating brain maps were created with Nilearn (v. 0.8.1; https://nilearn.github.io/stable/index.html).

### Reporting summary

Further information on research design is available in the Nature Portfolio Reporting Summary linked to this article.

## Data availability

Source data are provided as a Source Data file. The individual fMRI data generated in this study and necessary to reproduce the voxel-based and network-based results have been deposited in the Open Science Framework database under accession code https://osf.io/7nhsg. The individual pre-processed fMRI data are not publicly available due to size limitations but are available from the corresponding authors upon request. The group-level statistical brain maps (subsequent memory effects, memorability-corrected subsequent memory effects, voxel-based brain–behavior correlations of the encoding contrast, voxel-

based brain–behavior correlations of the subsequent memory effect contrast, functional connectivity networks with brain–behavior correlations, arousal-corrected subsequent memory effects, memorability effects) have been deposited on the NeuroVault database under the accession code http://neurovault.org/collections/14303/[92], and the full set of 60 ICs, as calculated from subsample 1, has been deposited on Figshare under the https://doi.org/10.6084/m9.figshare.c.6679262. Source data are provided in this paper.

## Code availability

The following publicly available software packages were used for preprocessing, analysis, and figure creation: Matlab (v. R2016b), SPM12 (v. 6685), MRTools' GLM Flex Fast 2, FreeSurfer (v. 4.5), the FMRIB Software Library, FSL MELODIC (v. 5.0.9), FSL dual regression (v. 5.0.9), Nilearn (v. 0.8.1), RStudio (2022), and the related R packages ggplot2 (v. 3.4.0), ggdist (v. 3.3.0), networkD3 (v. 0.4), nlme (v. 3.1–153), dplyr (v. 1.0.10), and oro.nifti (v. 0.11.4).

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

## Acknowledgements

This work has received funding from grants from the Swiss National Science Foundation (grant number 130080 to DJFQ and AP) and from the University of Basel, Switzerland.

## Author contributions

Conceptualization: L.G., D.C., D.J.F.Q., and A.P. Methodology: L.G., D.C., D.J.F.Q., and A.P. Software: L.G. and D.C. Validation: L.G. and D.C. Formal analysis: L.G. and D.C. Investigation: L.G. and D.C. Resources: L.G., D.C., D.J.F.Q., and A.P. Data curation: L.G. and D.C. Visualization: L.G. and D.C. Supervision: D.C., D.J.F.Q., and A.P. Project administration: L.G., D.C., D.J.F.Q., and A.P. Funding acquisition: D.J.F.Q. and A.P. Writing—original draft: L.G., D.C., and D.J.F.Q.. Writing—review & editing: L.G., D.C., D.J.F.Q., and A.P.

## Competing interests

The authors declare no competing interests.
