## [Peer Review File · Nature Communications]

Neurofunctional underpinnings of individual differences in visual episodic memory performanceReviewer #1 (Remarks to the Author):

Re: Neurofunctional underpinnings of individual differences in visual episodic memory performance

In this study the authors report results from a large number of participants (n=1485) who carried out a picture memory encoding task during fMRI scanning. The authors report a number of regions showing subsequent memory effects. They also report a subset of these regions that show inter-individual associations between the size of the subsequent memory effect and overall task performance. Finally, brain responses are decomposed into networks and the associations between network activity and subsequent memory effects are reported.

The strength of this study is the large number of participants. The results convincingly replicate previous findings about the regions that show subsequent memory effects – and do so in a very large sample. The main weakness of the study is that it feels exploratory and descriptive in nature. There was no theoretical framework provided to explain the motivation for the study – or summary of the hypotheses. Without this, it was unclear how the similarities and differences between the analyses should be interpreted.

The results seem to be broadly in agreement: a similar group of brain regions are more active for subsequently recognized items compared to subsequently forgotten items. Individuals who perform best on the task activate many of these regions more strongly during successful encoding. Networks that overlap with these regions also show greater activity during encoding. It is reassuring to see these effects, but I do not feel that it adds much to our understanding of the neural basis of memory processes.

More interesting are the regions which do not show consistent effects across the analyses. For example, some regions show subsequent memory effects but not inter-individual associations of the effect with memory. How could this be interpreted – and does it tell us anything novel about memory? Unfortunately, given the purely correlational nature of fMRI, I do not think it is possible to draw strong conclusions from such findings.

I really liked that the authors placed their results on NeuroVault. Overall, I felt that this was a solid study and that the results will be of interest to researchers in the field. However, I feel that the theoretical contribution of the study is modest and would be more suited to a more specialized journal.

Reviewer #2 (Remarks to the Author):

Geissmann et al. report a study of over 1400 adults who have performed a memory task that involves recalling pictures after having viewing them during fMRI data acquisition. The authors examine “subsequent memory” effects that are computed by contrasting trials in which the picture was subsequently recalled with trials in which the picture was not recalled. The authors also examined voxelwise brain behaviour correlations between (average?) activity during encoding and the number of recalled pictures for a given participant. Finally, expression of ICA networks were also correlated with inter-individual differences in performance. Some further remarks below.

1 . We know from prior work that there is greater power to detect within-subject fMRI effects than individual differences effects. Therefore the finding that the individual differences effects are a subset of the within-subject effects is not surprising. It is not really a fair comparison. One might examine instead whether there is a mapwise (i.e. spatial) correlation between brain-behavior and subsequent memory effects. This correlation might well be low, indicating a degree of independence. To find regions

associated with behavior that are not well predicted by the within-subject effect, one could examine the residuals of the mapwise regression just described, and map out areas of "low fit" between the two analyses.

2 . Subsequent memory effects may well be driven partly by item effects, since it is known that some images are much more "memorable"* than others. Brain-behavior effects are unlikely to be affected by item effects, since all items are included in the average (one assumes, but there is not detail in fact about what was used in the brain-behavior analysis). So this disconnect between levels of analysis would tend to reduce the similarity between the resultant maps. One might explicitly model memorability — and it would be easy to compute from the 1400 subjects by simply computing the number of times each image was recalled across the sample. This could then separate out memory encoding effects that cannot be explained by stimulus features and might sharpen the subsequent memory map—and in addition to providing a useful map of memorability effects.

3 . The authors have apparently (?) used the average estimated beta over all trials in the brain-behavior correlation. Have the authors considered examining the contrast (remembered - forgotten) to see whether better performers show a greater (or lesser) difference between subsequently remembered and subsequently forgotten items? There may be both trait effects (that, e.g. are similar for remembered and forgotten items) vs more transient effects that differ between remembered and forgotten items. So I think it worth examining how the use of contrast maps affect the brain-behavior analysis.

4 . I could not find any information about MRI scanning parameters. Also could find minimal information about experimental task timing (e.g. ISI) or how the images were projected and viewed by subjects. Also, more detail about how exactly the picture recall data was scored would be important.

5 . How reproducible are the brain-behavior effects? The recent paper by Marek et al.** suggests that such effects tend to be small and show large sample variance. It would be nice to see one of the resampling style analyses used by Marek et al. to establish the degree to which subsamples of the dataset tend to produce similar results and a low proportion of a sign flips.

*Khosla, A., Raju, A. S., Torralba, A., & Oliva, A. (2015). Understanding and predicting image memorability at a large scale. In Proceedings of the IEEE international conference on computer vision (pp. 2390-2398).

**Marek, S., Tervo-Clemmens, B., Calabro, F. J., Montez, D. F., Kay, B. P., Hatoum, A. S., ... & Dosenbach, N. U. (2022). Reproducible brain-wide association studies require thousands of individuals. *Nature*, 603(7902), 654-660.

Minor:

Line 66 — "In another words: It is ..."

Line 67 — "It is unclear to what extent brain regions that are essential for successful memory encoding" — the word "essential" is making a claim of necessity.

Line 97: This is not very clear: "This data-driven procedure allows taking into account the non-universality of FCNs across tasks and different populations of individuals".

Line 115: In the Intro and first line of Results, we read about pictures being freely recalled. But the reader is confused: how does one "recall" a picture? To help the reader, it is easy enough to explain briefly what it means to recall a picture in this study.

Line 184: "Since IC 54, in addition 184 to grey matter involvement, has a spatial distribution indicative of noise components 185 (large involvement of ventricles) (Fig. S8), we refrained from further interpreting IC 54." But is this not worthy of interpretation? Why does a network with ventricular involvement correlate with subsequent memory performance? Is it an error? Is it motion-related? etc.

"independent raters were responsible for the scoring to guarantee inter-rater validity."
– where is this reported?

Reviewer #3 (Remarks to the Author):

The current fMRI study sought to articulate the neural correlates of subsequent episodic memory using a large, single-site sample of participants (N=1,485). Specifically, the study presented participants with 72 affective images while undergoing fMRI followed by a test of written recall. The fMRI data was then analyzed using 1) standard group-level univariate analysis and 2) Independent Component Analysis (ICA) with a focus on which aspects of the measurements correlated with individuals' subsequent recall performance. The approach revealed a distributed, but non-identical set of brain regions whose activity/component loadings significantly correlated with performance.

There were several positive aspects of the study. In particular, the large sample size offers an unparalleled opportunity to explore individual differences in subsequent memory. The neural underpinnings of subsequent memory are of great theoretical and practical importance to range of memory researchers and clinicians, and so the topic and results are likely of interest to a broad readership. These positive aspects aside, there were also some significant issues that should be addressed.

First, the described dataset appears to offer the potential to address a large number of questions related to subsequent memory given the very large sample size. That being said, it was difficult to discern the primary set of hypotheses or conceptual questions being addressed in the current manuscript. For example, why was the main goal to relate ICA to the univariate approach? Are there any theoretical or mechanistic accounts that argue that the approaches should lead to different results? Are there any core dimensions of individual differences (e.g., demographics, idiosyncratic encoding strategies, sensitivity to affective stimuli) that were thought to give rise to the neural findings that could be explored? It seems as though there is quite a bit of potential in this dataset, but the analyses reported seemed to only catalogue basic findings rather than explore any central debates within the subsequent memory literature. This treatment of the data seems to be better suited for a publication which specializes in the release/description of new (and of potentially high value) datasets.

Second, it is unclear the degree to which the use of affective stimuli limits the generalizability of the results to subsequent memory more broadly. That is, is it possible that at least some of the results are due to the affective nature of the stimuli? For example, were more evocative images remembered a greater rate than less evocative images? Might the individual differences observed reflect the differential engagement of, e.g., arousal across participants? Perhaps a series of mediation analyses could help resolve how affective content did or did not influence the behavioral/neural measures of subsequent memory.

In addition to these significant issues, there were some minor issues as well:

-The methods section would benefit from the inclusion of more details. For example, how many scanning runs were there? How long were the runs (and the TRs)? There was reference to 'primacy and recency' items, but it was unclear what this meant. How long was each trial and how were the rating scales presented? Perhaps the use of a figure showing the trial structure would be helpful here.

-Relatedly, the description of the geometric figures was a little bit confusing. Were they presented 1 at a time or were they repeated? I assumed there must be some repetition as there were 25 of them, but 72 IAPS images, and a maximum of 2 IAPS images between shape trials.

-Perhaps this is related to my confusion above, but it is stated that "The difference between IAPS pictures and geometrical figures estimates... was used as a measure of task-related function responsivity of each IC". Does this mean that it is believed that there was no memory encoding taking place during the geometric shape trials or was something else meant?

-The lack of any negative subsequent memory effects is a bit surprising given their appearance in a number of past findings. Perhaps this could be speculated on some in the Discussion.

Reviewer 1

We thank the reviewer for pointing out the strengths and areas of improvements of our study and for providing helpful comments. In the meantime, we have included an additional 64 participants in the study and conducted further analyses regarding the effects of picture memorability, the comparison between different approaches and reproducibility. These new findings, coupled with a comprehensive revision of the entire manuscript, have significantly strengthened the paper.

In this study the authors report results from a large number of participants (n=1485) who carried out a picture memory encoding task during fMRI scanning. The authors report a number of regions showing subsequent memory effects. They also report a subset of these regions that show inter-individual associations between the size of the subsequent memory effect and overall task performance. Finally, brain responses are decomposed into networks and the associations between network activity and subsequent memory effects are reported.

The strength of this study is the large number of participants. The results convincingly replicate previous findings about the regions that show subsequent memory effects – and do so in a very large sample. The main weakness of the study is that it feels exploratory and descriptive in nature. There was no theoretical framework provided to explain the motivation for the study – or summary of the hypotheses. Without this, it was unclear how the similarities and differences between the analyses should be interpreted.

The theoretical framework and study rationale become clearer in the revised manuscript (see Introduction, e.g., lines 67-75). Further, we explicitly raised the specific research questions that we were able to address with the data of this large cohort: How does a classical group-based subsequent memory effect analysis of our data align with the findings from the meta-analysis on subsequent memory effects? In what ways do the results of our subsequent memory effect analysis change when controlling for memorability? What results emerge from a voxel-based brain-behavior correlation approach exploring brain activations related to inter-individual differences in memory performance, and how do these findings relate to the memorability-controlled subsequent memory effects? And, finally, what results emerge from a network-based approach investigating the neural correlates of inter-individual differences in memory performance? (Introduction, lines 119-128)

Apart from advancing our basic understanding of the neural correlates contributing to the variability in episodic memory performance among individuals, the present study could provide a foundation for future research aimed at relating individual biological characteristics to specific neural markers of episodic memory (Introduction, lines 128-132).

The results seem to be broadly in agreement: a similar group of brain regions are more active for subsequently recognized items compared to subsequently forgotten items. Individuals who perform best on the task activate many of these regions more strongly during successful encoding. Networks that overlap with these regions also show greater activity during encoding. It is reassuring to see these effects, but I do not feel that it adds much to our understanding of the neural basis of memory processes.

While group-based studies investigating subsequent memory effects have repeatedly revealed brain regions associated with memory, the interpretation of subsequent memory effects has been recently challenged since most of the studies have not controlled for the item memorability effect (i.e., that some pictures are inherently more memorable than others). The lack of accounting for memorability may pose a challenge to the interpretation of the previously reported subsequent memory effects as the portion of associated neural activity confounded by memorability may be substantial (Introduction, lines 69-75). We have

thus conducted an additional subsequent memory effect analysis controlling for item memorability. This analysis predominantly yielded a spatially similar activation pattern, which, however, was more focalized (i.e., including less voxels) and yielded lower t -values (Results, lines 162-179, Fig. 3). Moreover, this analysis detected an activation mainly located in the bilateral fusiform gyrus that was not present in the subsequent memory effect analysis that did not control for memorability (Fig. S4). These findings therefore contribute to the current debate (Discussion, lines 358-365).

Furthermore, our findings based on inter-individual differences in memory performance represent an important novelty as it was unknown to what extent brain regions that show subsequent memory effects are involved in explaining inter-individual differences. While this analysis revealed that several key regions, such as the hippocampus, orbitofrontal cortex, and posterior cingulate cortex, were correlated with inter-individual differences in memory performance, there were several regions detected in the subsequent memory analysis that were not correlated with inter-individual differences. To map out these regions, we performed an additional analysis examining the residuals of a regression analysis between the memorability-corrected subsequent memory analysis and the brain-behavior correlation analysis (Results, lines 206-211). These regions were mainly located in the lateral occipital cortex (Fig. 6). Importantly, the left and right inferior lateral occipital cortex were also not part of any of the FCNs correlated with memory performance. This area, which belongs to the visual associative cortex, has been linked to the initial encoding and subsequent memory of visual stimuli. Moreover, evidence from transcranial magnetic stimulation studies supports a causal role of this region in visual memory. Thus, while the lateral occipital cortex appears to play a role in successful visual memory encoding, inter-individual differences in encoding-related brain activation in this region did not contribute to memory variability in the present study (Discussion, lines 376-388).

We now also provide a resampling analysis, demonstrating the robustness of our voxel-based brain-behavior correlations and indicating that indeed large samples between 500 and 1000 participants are needed for this kind of analysis (Results, lines 188-197; Fig. 5; Discussion, lines 457-462). Finally, the network-based approach revealed that the right cerebellar hemisphere, a region that was neither detected in the memorability-corrected subsequent memory effect analysis nor in the voxel-based brain-behavior correlation analysis, was part of a functional connectivity network explaining inter-individual differences (Discussion, lines 442-444).

More interesting are the regions which do not show consistent effects across the analyses. For example, some regions show subsequent memory effects but not inter-individual associations of the effect with memory. How could this be interpreted – and does it tell us anything novel about memory? Unfortunately, given the purely correlational nature of fMRI, I do not think it is possible to draw strong conclusions from such findings.

We agree that based on fMRI data alone we cannot draw causal conclusions about regions differently involved in the subsequent memory effect analysis and the analyses based in inter-individual differences in memory performance. The findings of the present study, however, may offer a basis for further mechanistic studies in animals, or in humans. In this context, we now discuss that for the lateral occipital cortex, which was related to successful memory encoding but not to inter-individual differences in episodic memory performance, evidence from transcranial magnetic stimulation studies supports a causal role of this region in visual memory. Thus, while the lateral occipital cortex appears to play a role in successful visual memory encoding, inter-individual differences in encoding-related brain activation in this region did not contribute to memory variability in the present study (Discussion, lines 373-388).

I really liked that the authors placed their results on NeuroVault. Overall, I felt that this was a solid study and that the results will be of interest to researchers in the field. However, I feel that the theoretical contribution of the study is modest and would be more suited to a more specialized journal.

We understand your concerns about the theoretical contribution of our work. In response to your comments, we have endeavored to enhance this aspect in the revised manuscript by performing additional analyses and by strengthening the introduction and discussion. We trust that these improvements emphasize the unique contributions of our study and enhance its suitability for a publication with a broader readership.

Reviewer 2

We thank the reviewer for pointing out the strengths and areas of improvements of our study and for providing helpful comments. In the meantime we have included an additional 64 participants in the study and conducted further analyses regarding the effects of picture

memorability, the comparison between different approaches and reproducibility. These new findings, coupled with a comprehensive revision of the entire manuscript, have significantly strengthened the paper.

Geissmann et al. report a study of over 1400 adults who have performed a memory task that involves recalling pictures after having viewing them during fMRI data acquisition. The authors examine “subsequent memory” effects that are computed by contrasting trials in which the picture was subsequently recalled with trials in which the picture was not recalled. The authors also examined voxelwise brain behaviour correlations between (average?) activity during encoding and the number of recalled pictures for a given participant. Finally, expression of ICA networks were also correlated with inter-individual differences in performance. Some further remarks below.

1 . We know from prior work that there is greater power to detect within-subject fMRI effects than individual differences effects. Therefore, the finding that the individual differences effects are a subset of the within-subject effects is not surprising. It is not really a fair comparison. One might examine instead whether there is a mapwise (i.e. spatial) correlation between brain-behavior and subsequent memory effects. This correlation might well be low, indicating a degree of independence. To find regions associated with behavior that are not well predicted by the within-subject effect, one could examine the residuals of the mapwise regression just described, and map out areas of "low fit" between the two analyses.

Thank you for this input. Indeed, the suggested analyses are of great value. We have performed a regression analysis between the brain-behavior and the memorability-corrected subsequent memory effect analysis (Fig. S9) and used the residuals to map out regions where the brain-behavior correlations were lower than expected from the memorability-corrected subsequent memory effects (Results, lines 206-211). These regions were mainly located in the lateral superior and inferior occipital cortex (Fig. 6).

2 . Subsequent memory effects may well be driven partly by item effects, since it is known that some images are much more “memorable” than others. Brain-behavior effects are unlikely to be affected by item effects, since all items are included in the average (one assumes, but there is not detail in fact about was used in the brain-behavior analysis). So this disconnect between levels of analysis would tend to reduce the similarity between the resultant maps. One might explicitly model memorability — and it would be easy to compute from the 1400 subjects by simply computing the number of times each image was recalled across the sample. This could then separate out memory encoding effects that cannot be explained by stimulus features and might sharpen the subsequent memory map--and in addition to providing a useful map of memorability effects.

We fully agree with the reviewer and have modelled memorability accordingly. We then performed the subsequent memory effect analysis while statistically controlling for memorability. Whereas to a large degree similar brain regions emerged, the extent of significant voxels and the t -values were smaller after controlling for memorability (Results, lines 162-173; Fig. 3). Moreover, this analysis detected an activation mainly located in the bilateral fusiform gyrus that was not present in the subsequent memory effect analysis that did not control for memorability (Fig. S4). We also created maps on positive and negative memorability effects (Results, lines 174-175; Supplementary Materials, Methods, S1; Fig. S6 and S7).

3 . The authors have apparently (?) used the average estimated beta over all trials in the brain-behavior correlation. Have the authors considered examining the contrast (remembered - forgotten) to see whether better performers show a greater (or lesser) difference between subsequently remembered and subsequently forgotten items? There may be both trait effects (that, e.g. are similar for remembered and forgotten items) vs more transient effects that different between remembered and forgotten items. So I think it worth examining how the use of contrast maps effect the brain-behavior analysis.

Yes, we have used the average estimated beta over all trials in the brain-behavior correlation (now described in the Methods, lines 658-662). In addition to the brain-behavior correlation model where responsivity to pictures was based on the picture encoding contrast, we now also constructed a model where the brain variable captured subsequent memory effects (Results, lines 212-221; Methods, line 664-665). Whereas this analysis did not reveal any significant positive correlations, we observed negative correlations with episodic memory performance in a few regions, most prominently in the lateral occipital cortex (Fig. S10), indicating that better performers show reduced subsequent memory effects in these regions as compared to lower performers. Of note, the lateral occipital cortex showed subsequent memory effects (Fig. 3), but a lack of brain-behavior correlation using the picture encoding contrast (Fig. 6).

4 . I could not find any information about MRI scanning parameters. Also could find minimal information about experimental task timing (e.g. ISI) or how the images were projected and viewed by subjects. Also, more detail about how exactly the picture recall data was scored would be important.

We have updated the Methods section to include all relevant fMRI parameters and task details (Methods, lines 487-543).

5 . How reproducible are the brain-behavior effects? The recent paper by Marek et al.** suggests that such effects tend to be small and show large sample variance. It would be nice to see one of the resampling style analyses used by Marek et al. to establish the degree to which subsamples of the dataset tend to produce similar results and a low proportion of a sign flips.

Thank you for this input. To test the robustness of the voxel-based brain-behavior correlations based on the picture encoding contrast, we applied the suggested resampling procedure. This procedure consisted in estimating the effect size of brain-behavior correlations for sample sizes ranging from 26 to 1,000 participants. Five thousand random samples were selected for each sample size. This analysis demonstrated a similar trend in effect sizes as the one reported by Marek et al. (2022): at small sample sizes, the association (i.e., brain-behavior correlation) was not reproducible and exhibited a lot of variability and sign changes. The effect size converged at larger sample sizes and stabilized for sample sizes greater than 500 participants (Fig. 5). This analysis has been added to the manuscript (Results, lines 188-197; Methods, lines 670-684).

*Khosla, A., Raju, A. S., Torralba, A., & Oliva, A. (2015). Understanding and predicting image memorability at a large scale. In *Proceedings of the IEEE international conference on computer vision* (pp. 2390-2398).

Will be added to the final version of the manuscript

**Marek, S., Tervo-Clemmens, B., Calabro, F. J., Montez, D. F., Kay, B. P., Hatoum, A. S., ... & Dosenbach, N. U. (2022). Reproducible brain-wide association studies require thousands of individuals. *Nature*, 603(7902), 654-660.

Has been added to the revised version of the manuscript

Minor comments:

Line 66 — “In another words: It is ...”

Corrected. Lines, 79-80

Line 67 — “It is unclear to what extent brain regions that are essential for successful memory encoding” — the word “essential” is making a claim of necessity.

Changed to: “It is unclear to what extent brain regions related to successful memory encoding...” Line 80

Line 97: This is not very clear: “This data-driven procedure allows taking into account the non-universality of FCNs across tasks and different populations of individuals”.

Changed to: “As brain activity differs between tasks and between populations of individuals, using this data-driven procedure instead of a template-based one circumvents violating the assumption of across-task- and across-population equality in the spatial topology of FCNs 28–30” Lines 111-114

Line 115: In the Intro and first line of Results, we read about pictures being freely recalled. But the reader is confused: how does one “recall” a picture? To help the reader, it is easy enough to explain briefly what it means to recall a picture in this study.

The description of the picture recall task has been made clearer in the Introduction (lines 117-119; Results, lines 147-149 and in the Methods, lines 521-529).

“independent raters were responsible for the scoring to guarantee inter-rater validity.” — where is this reported?”

The description of the picture rating procedure and information regarding inter-rater reliability has been added to the Methods (lines 521-529).

Line 184: “Since IC 54, in addition 184 to grey matter involvement, has a spatial distribution indicative of noise components 185 (large involvement of ventricles) (Fig. S8), we refrained from further interpreting IC 54.” But is this not worthy of interpretation? Why does a network with ventricular involvement correlate with subsequent memory performance? Is it an error? Is it motion-related? etc.

We agree and included IC 54 in the Results section, providing further information to our readers (Results, lines 307-315). We then offer our interpretation in the Discussion (lines 421-426): In contrast to the other eight ICs, IC 54 stands out with combining gray matter and prominent spatial characteristics indicative of noise components. The latter include a fragmented appearance, large involvement of ventricles, and ring-like stripes near the edges of the field of view. An involvement of the insula, temporal gyri and hippocampus may have fostered this IC to have brain-behavior correlations despite these noise components.

Reviewer 3

We thank the reviewer for pointing out the strengths and areas of improvements of our study and for providing helpful comments. In the meantime we have included an additional 64 participants in the study and conducted further analyses regarding the effects of picture memorability, the comparison between different approaches and reproducibility. These new findings, coupled with a comprehensive revision of the entire manuscript, have significantly strengthened the paper.

The current fMRI study sought to articulate the neural correlates of subsequent episodic memory using a large, single-site sample of participants (N=1,485). Specifically, the study presented participants with 72 affective images while undergoing fMRI followed by a test of written recall. The fMRI data was then analyzed using 1) standard group-level univariate analysis and 2) Independent Component Analysis (ICA) with a focus on which aspects of the measurements correlated with individuals' subsequent recall performance. The approach revealed a distributed, but non-identical set of brain regions whose activity/component loadings significantly correlated with performance.

There were several positive aspects of the study. In particular, the large sample size offers an unparalleled opportunity to explore individual differences in subsequent memory. The neural underpinnings of subsequent memory are of great theoretical and practical importance to range of memory researchers and clinicians, and so the topic and results are likely of interest to a broad readership. These positive aspects aside, there were also some significant issues that should be addressed.

First, the described dataset appears to offer the potential to address a large number of questions related to subsequent memory given the very large sample size. That being said, it was difficult to discern the primary set of hypotheses or conceptual questions being addressed in the current manuscript. For example, why was the main goal to relate ICA to the univariate approach? Are there any theoretical or mechanistic accounts that argue that the approaches should lead to different results? Are there any core dimensions of individual differences (e.g., demographics, idiosyncratic encoding strategies, sensitivity to affective stimuli) that were thought to give rise to the neural findings that could be explored? It seems as though there is quite a bit of potential in this dataset, but the analyses reported seemed to only catalogue basic findings rather than explore any central debates within the subsequent memory literature. This treatment of the data seems to be better suited for a publication which specializes in the release/description of new (and of potentially high value) datasets.

Thank you for this input. The theoretical framework and study rationale become clearer in the revised manuscript (see Introduction, e.g., lines 67-75). Further, we explicitly raised the specific research questions that we were able to address with the data of this large cohort: How does a classical group-based subsequent memory effect analysis of our data align with the findings from the meta-analysis on subsequent memory effects? In what ways do the results of our subsequent memory effect analysis change when controlling for memorability? What results emerge from a voxel-based brain-behavior correlation approach exploring brain activations related to inter-individual differences in memory performance, and how do these findings relate to the memorability-controlled subsequent memory effects? And, finally, what results emerge from a network-based approach investigating the neural correlates of inter-individual differences in memory performance? The data from our large sample and the combined use of group-based and inter-individual approaches allowed us to answer these open questions. (Introduction, lines 119-128)

With regard to subsequent memory effects, we have performed an additional analysis that provides data regarding a current debate: While group-based studies investigating subsequent memory effects have repeatedly revealed brain regions associated with memory, the interpretation of subsequent memory effects has been recently challenged since most of the studies have not controlled for the item memorability effect (i.e., that some pictures are inherently more memorable than others). The lack of accounting for memorability may pose a challenge to the interpretation of the previously reported subsequent memory effects as the portion of associated neural activity confounded by memorability may be substantial. We have thus conducted an additional subsequent memory effect analysis controlling for item memorability. This analysis predominantly yielded a spatially similar activation pattern, which, however, was more focalized (i.e., including less voxels) and yielded lower *t*-values (Fig. 3). Moreover, this analysis detected an activation mainly located in the bilateral fusiform gyrus that was not present in the subsequent memory effect analysis that did not control for memorability (Fig. S4). (Results, lines 162-175)

Second, it is unclear the degree to which the use of affective stimuli limits the generalizability of the results to subsequent memory more broadly. That is, is it possible that at least some of the results are due to the affective nature of the stimuli? For example, were more evocative images remembered a greater rate than less evocative images? Might the individual differences observed reflect the differential engagement of, e.g., arousal across participants? Perhaps a series of mediation analyses could help resolve how affective content did or did not influence the behavioral/neural measures of subsequent memory.

Thank you for bringing up this important point. When correcting for picture arousal, we found a spatially similar activation pattern, which, however, was more focalized (i.e., including less voxels) and yielded lower *t*-values (Fig. S8). This analysis has been added to the manuscript (Results, lines 176-179; Methods, lines 633-647). Furthermore, since the affective content of pictures along with other features, such as aesthetics, contribute to picture memorability, we now provide an additional analysis of the subsequent memory effect controlling for picture memorability effects (see point above).

In addition to these significant issues, there were some minor issues as well:

The methods section would benefit from the inclusion of more details. For example, how many scanning runs were there? How long were the runs (and the TRs)? There was reference to 'primacy and recency' items, but it was unclear what this meant. How long was each trial and how were the rating scales presented? Perhaps the use of a figure showing the trial structure would be helpful here.

We have updated the Methods section to include all relevant fMRI parameters and task details (Methods, lines 487-543).

Relatedly, the description of the geometric figures was a little bit confusing. Were they presented 1 at a time or were they repeated? I assumed there must be some repetition as there were 25 of them, but 72 IAPS images, and a maximum of 2 IAPS images between shape trials.

There were 24 scrambled pictures with geometrical figures, 24 positive IAPS pictures, 24 negative IAPS pictures and 24 neutral IPAS pictures. We now provide a clearer description related to the geometric figures in the Methods section (Methods, lines 495-504 and 509-511).

Perhaps this is related to my confusion above, but it is stated that “The difference between IAPS pictures and geometrical figures estimates... was used as a measure of task-related function responsivity of each IC”. Does this mean that it is believed that there was no memory encoding taking place during the geometric shape trials or was something else meant?

For the voxel-based and network-based brain-behavior correlations we have used the contrast between the neural activity during viewing IAPS pictures and the neural activity during viewing scrambled pictures (with geometrical figures on top of them). This contrast yields neural activity related to viewing real pictures. Whereas there might be encoding-related brain activity also for geometrical figures, we have shown previously that this contrast contains activations in brain regions typically involved in successful memory encoding of pictures (Fastenrath et al., J Neurosci. 2014). We have now made a clearer description in the Methods (lines, 658-662).

The lack of any negative subsequent memory effects is a bit surprising given their appearance in a number of past findings. Perhaps this could be speculated on some in the Discussion.

In fact, we did find negative subsequent memory effects in the central opercular cortex, Heschl's gyrus, precuneus, right frontal pole, right intracalcarine and lingual gyrus, juxtapositional junction, and the precentral gyrus (Results, lines 159-162; Fig. S2). These findings are in line with the results of the meta-analysis by Kim et al. (2011). We also detected memorability-controlled negative subsequent memory effects in regions similar to the classical negative subsequent memory effects (Results, lines 170-173; Fig. S5). Regions that did not show any significant negative effects when controlling for memorability were the intracalcarine gyrus, lingual gyrus, and precentral gyrus.

Reviewer #1 (Remarks to the Author):

Re: Neurofunctional underpinnings of individual differences in visual episodic memory performance

In this revision, the authors include a number of additional analyses which address specific concerns raised by the reviewers. These additions are helpful and considerably improve the quality of the study. I particularly liked the new analysis which controls for memorability and the new analysis with the resampled datasets.

In my review, I raised a concern that the study felt exploratory and descriptive in nature and it lacked a theoretical framework. The authors have responded that they have addressed this in their revision and that they "explicitly raised the specific research questions". These are:

"How does a classical group-based subsequent memory effect analysis of our data align with the findings from the meta-analysis on subsequent memory effects?"

In what ways do the results of our subsequent memory effect analysis change when controlling for memorability?"

What results emerge from a voxel-based brain-behavior correlation approach exploring brain activations related to inter-individual differences in memory performance, and how do these findings relate to the memorability-controlled subsequent memory effects?"

What results emerge from a network-based approach investigating the neural correlates of inter-individual differences in memory performance?"

It is useful to have these spelled out. Nevertheless, I still feel that these are "exploratory and descriptive" – they refer to results emerging from analyses and to examining how the results of different analyses relate or align with each other. The research questions address issues that are largely methodological in nature and of interest to specialists working in the field, but do not clearly address larger theoretical issues. In the manuscript, the authors present a set of results from each of the analyses and these results are qualitatively compared to identify similarities and differences – but a deeper discussion of the implications of these findings is lacking.

I have no doubt that the findings reported in the paper will be of interest to the large number of cognitive neuroscientists who use fMRI to investigate memory. I am sure that the manuscript will be well-cited; it reflects the most comprehensive report of subsequent memory effects in recognition memory using fMRI to date. However, for me, it does little to advance our understanding of how memory processes operate in the brain.

Reviewer #2 (Remarks to the Author):

This is a very thorough analysis of a large memory dataset and the authors have nicely addressed points raised in the previous review.

Reviewer #3 (Remarks to the Author):

I thank the authors for their substantial revisions which I believe have strengthened the manuscript considerably. Based on these revisions, I have a few follow-up questions. The updated framing highlights three main motivations for the current study: compare variation in the neural subsequent memory effect across individuals to the average effect, compare a large single-study sample to existing meta-analytic results, and account for picture memorability as a potential confound. I believe that the first line of motivation is the most important and the study seems well-positioned to speak to it (and indeed the title and abstract highlight this motivation). The motivation for the other two points feels less clear. For example, what is the benefit of having a large, within-sample study compared to a meta-analysis? It is stated that it is lacking, but the

benefit should be explicitly stated. Similarly, understanding the influence of picture memorability seems important, but it is unclear why it should be a central focus (rather than an important methodological consideration) for this particular study. Is the confound a particular concern for the meta-analysis? It would seem like using visual and verbal study material in that analysis might insulate them from the concern, but perhaps not? Conversely, is there something about memorability that can be gleaned from this unique dataset? For example, are there robust individual differences in memorability effects that are potentially interesting and/or complementary to the other results described?

By way of a follow up on a minor concern raised before, it is still not stated how long each trial is (including responses to the image) and it still might be helpful to have a figure depicting an example trial and example IAPS/geometric shape stimulus.

Per another minor point, I appreciate the authors attempt to clarify the ordering constraints of the geometric shapes and IAPS images. However, my basic confusion is around the constraints that: 1) there are at most 2 IAPS images for every geometric shape ("scrambled pictures with simple geometrical figures were presented in such a way that a maximum of two IAPS pictures were presented in succession") and 2) the ratio of IAPS images to shapes is substantially greater than 2:1 (i.e., it was 72:24). Some clarification would be helpful and the only real concern would be if the geometric shapes were repeated as that might confound memory recognition effects with visual category.

A new minor point raised by revisions concerns the claim: "...there were several brain regions related to successful memory encoding that did not explain inter-individual differences in episodic memory performance. These regions were mainly located in the lateral occipital cortex. Importantly, the left and right inferior lateral occipital cortex were also not part of any of the FCNs correlated with memory performance." However, unless I am missing something, Figure S10 implies a relationship, albeit negative, which presumably is a slightly different (but potentially very interesting) interpretation.

Reviewer #1

Re: Neurofunctional underpinnings of individual differences in visual episodic memory performance

In this revision, the authors include a number of additional analyses which address specific concerns raised by the reviewers. These additions are helpful and considerably improve the quality of the study. I particularly liked the new analysis which controls for memorability and the new analysis with the resampled datasets. In my review, I raised a concern that the study felt exploratory and descriptive in nature and it lacked a theoretical framework. The authors have responded that they have addressed this in their revision and that they “explicitly raised the specific research questions”. These are: “How does a classical group-based subsequent memory effect analysis of our data align with the findings from the meta-analysis on subsequent memory effects? In what ways do the results of our subsequent memory effect analysis change when controlling for memorability? What results emerge from a voxel-based brain-behavior correlation approach exploring brain activations related to inter-individual differences in memory performance, and how do these findings relate to the memorability-controlled subsequent memory effects? What results emerge from a network-based approach investigating the neural correlates of inter-individual differences in memory performance?” It is useful to have these spelled out. Nevertheless, I still feel that these are “exploratory and descriptive” – they refer to results emerging from analyses and to examining how the results of different analyses relate or align with each other. The research questions address issues that are largely methodological in nature and of interest to specialists working in the field, but do not clearly address larger theoretical issues. In the manuscript, the authors present a set of results from each of the analyses and these results are qualitatively compared to identify similarities and differences – but a deeper discussion of the implications of these findings is lacking. I have no doubt that the findings reported in the paper will be of interest to the large number of cognitive neuroscientists who use fMRI to investigate memory. I am sure that the manuscript will be well-cited; it reflects the most comprehensive report of subsequent memory effects in recognition memory using fMRI to date. However, for me, it does little to advance our understanding of how memory processes operate in the brain.

Response: We are pleased that Reviewer #1 qualifies our study as the most comprehensive report of subsequent memory effects to date and that this Reviewer also recognizes the contribution of the additional analyses to enhancing the quality of the study. We understand the perceived exploratory nature of our study; however, we argue that our analysis of the brain regions and networks linked to individual differences in memory performance is novel and marks a significant advancement in understanding memory processes. Importantly, our results also provide a novel foundation for downstream mechanistic and genetic studies, paving the way for a deeper understanding of memory processes.

In response to the feedback, we now discuss the implications more explicitly in the last paragraph of the discussion (lines 452-459): “In conclusion, our study identifies the key brain regions and networks related to individual differences in visual episodic memory performance. Notably, we found that certain regions, pivotal at the group-level, do not correlate with individual performance. These insights bear significant implications for research striving to link individual neurofunctional signals with psychological traits, or with genetic, epigenetic, or metabolomic profiles. Research of this nature would benefit from the selection of neurofunctional signals that are related to individual differences in memory performance, rather than those that emerge from group-level analyses.”

Reviewer #2

This is a very thorough analysis of a large memory dataset and the authors have nicely addressed points raised in the previous review.

Response: We are thankful to Reviewer #2 for the constructive comments and suggestions for additional analyses offered in the initial review, which significantly enhanced the quality of our work. We are pleased that our revisions have been well received.

Reviewer #3

I thank the authors for their substantial revisions which I believe have strengthened the manuscript considerably.

Response: We greatly appreciate Reviewer #3's acknowledgement of our efforts in strengthening the manuscript through substantial revisions. We found this reviewer's initial comments invaluable in guiding these enhancements and we are gratified to learn that our revisions have indeed improved the manuscript significantly.

Based on these revisions, I have a few follow-up questions.

The updated framing highlights three main motivations for the current study: compare variation in the neural subsequent memory effect across individuals to the average effect, compare a large single-study sample to existing meta-analytic results, and account for picture memorability as a potential confound. I believe that the first line of motivation is the most important and the study seems well-positioned to speak to it (and indeed the title and abstract highlight this motivation). The motivation for the other two points feels less clear. For example, what is the benefit of having a large, within-sample study compared to a meta-analysis? It is stated that it is lacking, but the benefit should be explicitly stated.

Response: We appreciate the reviewer's additional helpful comments in this round of review. To address the first point, we have integrated the following argument in the Introduction (lines 91-93): "Comparing the results from our large single-center study with those of the meta-analysis serves to establish the validity and robustness of our study. Additionally, such alignment helps corroborate and strengthen the overall findings of the meta-analysis."

Similarly, understanding the influence of picture memorability seems important, but it is unclear why it should be a central focus (rather than an important methodological consideration) for this particular study. Is the confound a particular concern for the meta-analysis? It would seem like using visual and verbal study material in that analysis might insulate them from the concern, but perhaps not? Conversely, is there something about memorability that can be gleaned from this unique dataset? For example, are there robust individual differences in memorability effects that are potentially interesting and/or complementary to the other results described?

Response: Indeed, the confound is a concern for the meta-analysis, as a memorability effect is not only observed for visual material but also for verbal material (Madan, C. R. (2021). Psychonomic Bulletin & Review, 28(2), 583-595). In light of this, we have adjusted our terminology to use "item memorability" instead of "picture memorability" (lines 95, 97) and added the reference regarding the memorability effect for verbal material. Given the significant proportion of neural activity that may be confounded by memorability, we deemed it essential to control the subsequent memory effect for this variable.

Regarding your question about whether something could be gleaned from the memorability results: The brain map illustrating the group-based positive memorability effects displays a robust activation pattern in regions associated with memory. This pattern aligns closely with those observed in the memorability-controlled subsequent memory effect analysis, a detail we've now included in the results section (lines 175-178). However, given that our study wasn't explicitly designed to examine all facets of memorability and considering memorability wasn't our primary focus, we've chosen not to delve deeper into this particular aspect.

By way of a follow up on a minor concern raised before, it is still not stated how long each trial is (including responses to the image) and it still might be helpful to have a figure depicting an example trial and example IAPS/geometric shape stimulus.

Response: We apologize for the omission of the timing details for a complete trial. To clarify, the total trial duration was between 12 and 15 s. This time frame included a 500-ms fixation cross, a 2.5-s picture presentation, and then a variable intertrial period ranging from 9 to 12 s (allowing for a 3-s jitter for stimulus onset) during which the pictures were rated. We have added this information to the Methods section (lines 497-500, 505-506). Moreover, following your suggestion, we have included a supplementary figure (Fig. S20) to illustrate an example trial.

Per another minor point, I appreciate the authors attempt to clarify the ordering constraints of the geometric shapes and IAPS images. However, my basic confusion is around the constraints that: 1) there are at most 2 IAPS images for every geometric shape ("scrambled pictures with simple geometrical figures were presented in such a way that a maximum of two IAPS pictures were presented in succession") and 2) the ratio of IAPS images to shapes is substantially greater than 2:1 (i.e., it was 72:24). Some clarification would be helpful and the only real concern would be if the geometric shapes were repeated as that might confound memory recognition effects with visual category.

Response: This concern can be ruled out, and we apologize for not having detailed this upfront. For the 24 scrambled pictures we have used 24 distinct geometrical figures. There was no repetition of items of this picture category. We have added this information to the Methods section (lines 487, 494-495).

A new minor point raised by revisions concerns the claim: "...there were several brain regions related to successful memory encoding that did not explain inter-individual differences in episodic memory performance. These regions were mainly located in the lateral occipital cortex. Importantly, the left and right inferior lateral occipital cortex were also not part of any of the FCNs correlated with memory performance." However, unless I am missing something, Figure S10 implies a relationship, albeit negative, which presumably is a slightly different (but potentially very interesting) interpretation.

Response: The negative relationship depicted in Figure S10 corresponds to the correlation between memory performance and the subsequent memory effect, not the correlation between memory performance and brain responsivity to picture encoding. We understand the potential for confusion and have clarified this point in the Discussion section of our manuscript (lines 366-367). Indeed, the negative correlation between memory performance and the subsequent memory effect in this region is interesting. Nonetheless, considering the speculative nature of any potential interpretation, we have chosen not to delve into it.